# 🦦 FERRET-UI 2: MASTERING UNIVERSAL USER INTERFACE UNDERSTANDING ACROSS PLATFORMS

**Zhangheng Li**[†]**, Keen You, Haotian Zhang, Di Feng, Harsh Agrawal, Xiujun Li,**
**Mohana Prasad Sathya Moorthy, Jeff Nichols, Yinfei Yang, Zhe Gan**
Apple
zoharli@utexas.edu, {yinfeiy,zhe.gan}@apple.com

## ABSTRACT

Building a generalist model for user interface (UI) understanding is challenging due to various foundational issues, such as platform diversity, resolution variation, and data limitation. In this paper, we introduce **Ferret-UI 2**, a multimodal large language model (MLLM) designed for universal UI understanding across a wide range of platforms, including iPhone, Android, iPad, Webpage, and AppleTV. Building on the foundation of Ferret-UI, Ferret-UI 2 introduces three key innovations: support for multiple platform types, high-resolution perception through adaptive scaling, and advanced task training data generation powered by GPT-4o with set-of-mark visual prompting. These advancements enable Ferret-UI 2 to perform complex, user-centered interactions, making it highly versatile and adaptable for the expanding diversity of platform ecosystems. Extensive empirical experiments on referring, grounding, user-centric advanced tasks (comprising 9 subtasks × 5 platforms), GUIDE next-action prediction dataset, and GUI-World multi-platform benchmark demonstrate that Ferret-UI 2 significantly outperforms Ferret-UI, and also shows strong cross-platform transfer capabilities.

## 1 INTRODUCTION

User interfaces (UIs) are central to human-computer interaction, shaping how users interact with digital systems. The complexity of UIs has evolved with the proliferation of platforms such as smartphones, tablets, web platforms, and smart TVs. Despite this increasing diversity, many current approaches to UI understanding and interaction (Hong et al., 2023; Wang et al., 2024b; Kapoor et al., 2024), particularly those in multi-platform ecosystems, face limitations.

One prominent effort in this space is Ferret-UI (You et al., 2024), which has advanced the field of referring and grounding UIs. However, though taking an any-resolution approach (Liu et al., 2024a), Ferret-UI is constrained by a fixed grounding resolution (*i.e.*, 336×672 and 672×336), and focuses on single-type platforms (*i.e.*, mobile devices including iPhone and Android), limiting its applicability in the context of today's highly diverse platform landscape. For example, as illustrated in Figure 1, one notable difference among these four exemplified platforms is resolution, the native resolution of iPhone differs from that of iPad, Web UI, and also AppleTV, directly applying Ferret-UI across these platforms presents significant challenges. Another major challenge is the lack of platform-specific, high-quality data, given different platforms. Though Ferret-UI's approach to training data generation can be extended to these platforms, it primarily relies on text-based GPT-4 prompting, where bounding boxes are represented in a purely textual format. This absence of visual input and spatial relationships between UI elements diminishes the quality of training data, which in turn limits the performance and effectiveness of the resulting model.

To address these limitations, we introduce **Ferret-UI 2**, a multimodal large language model (MLLM) designed to understand diverse UI screens and respond to user intent through single-step interactions across multiple platforms. Building on the foundation of Ferret-UI (You et al., 2024), Ferret-UI 2 significantly enhances UI perception and user interaction capabilities via three key advancements: (*i*) multi-platform support, (*ii*) dynamic high-resolution image encoding, and (*iii*) high-quality multimodal training data generation.

---

[†]University of Texas at Austin. Work done during an internship at Apple.

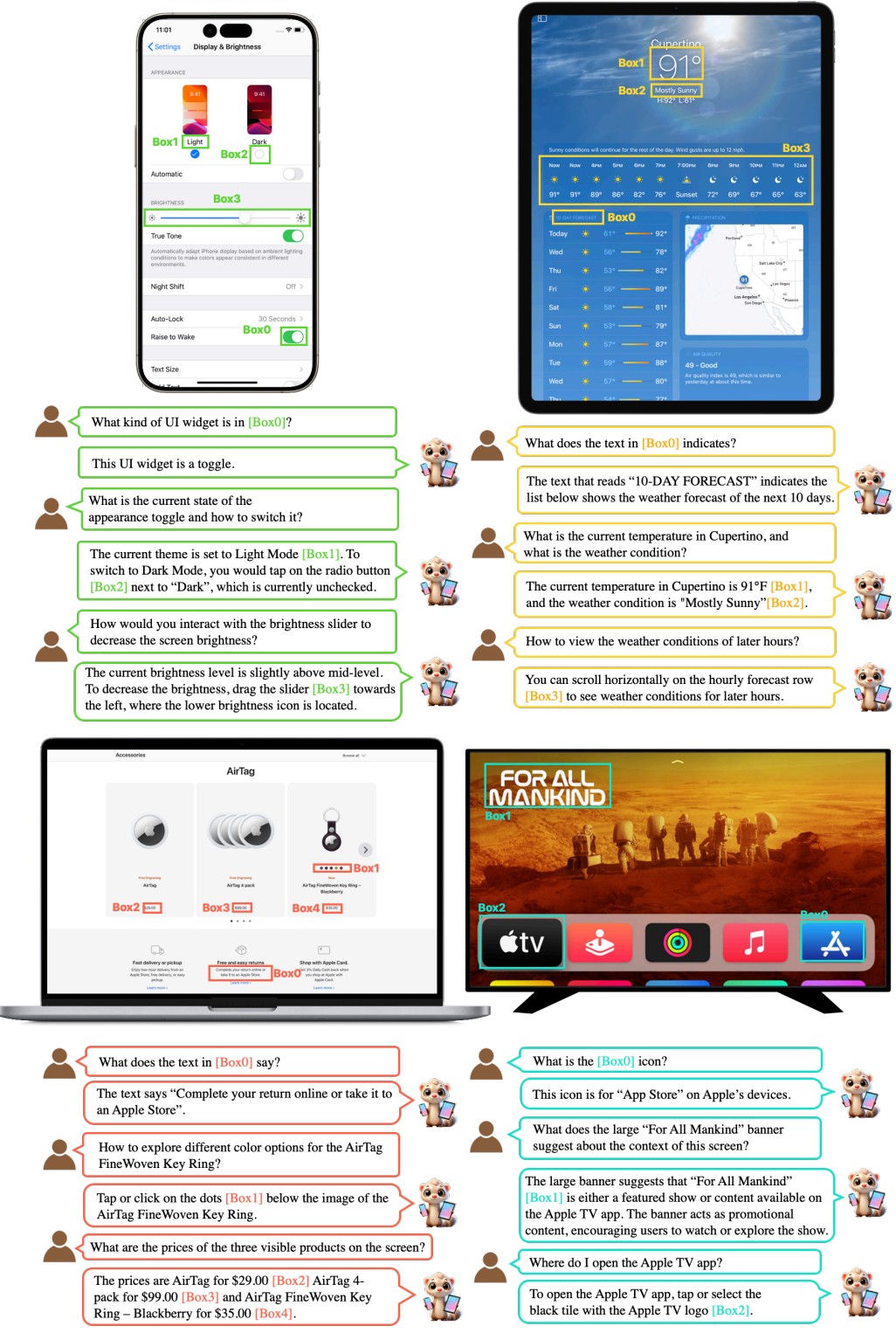

Figure 1: Real examples of a single Ferret-UI 2 model interacting with four different platforms (iPhone, iPad, Webpage, and AppleTV) for UI understanding. Refer to Appendix H for more examples including multi-step interactions.

First, Ferret-UI 2 extends its compatibility beyond mobile platforms (iPhone and Android), incorporating additional platforms like tablets, webpages, and smart TVs. Figure 1 illustrates visual examples of Ferret-UI 2 interacting with users across four typical screen types. This multi-platform support enables broader applicability and allows the system to seamlessly scale across a variety of user environments.

Second, Ferret-UI 2 supports high-resolution image encoding via the any-resolution method (Liu et al., 2024a; Zhang et al., 2024c). However, going beyond that, we introduce an enhanced *adaptive gridding* approach to maintain perception capabilities at the original resolution of the UI screenshot, ensuring more accurate recognition of visual elements. By leveraging human-collected bounding box annotations, we enhance referring and grounding precision, which allows for a more detailed understanding of UI components and their relationships.

Third, Ferret-UI 2 leverages high-quality training data for both elementary and advanced tasks. For elementary tasks, we convert simple referring and grounding data into conversations, allowing the model to develop a fundamental understanding of diverse UI screens. For advanced tasks, which focus on user-centered, free-form conversations, we replace the text-based GPT-4 prompting (where bounding boxes are described only in text) with GPT-4o using set-of-mark visual prompting (Yang et al., 2023) for training data generation. This approach enhances spatial understanding of UI elements, resulting in higher-quality training data. Additionally, unlike previous methods that use straightforward instructions such as "click on [bbox location]", Ferret-UI 2 performs single-step user-centered interactions. For example, when given a command like "please confirm submission", the system understands and executes the intended action, rather than simply following mechanical click instructions. Overall, our contributions are summarized as follows.

- We present Ferret-UI 2, a multimodal LLM that sets itself apart from previous efforts by supporting a broader range of platforms, including iPhone, Android, iPad, Webpage, and AppleTV. We upgrade Ferret-UI across multiple fronts, including better instruction-tuning data for model training, high-resolution image encoding for enhanced performance, and new referring and grounding benchmarks tailored for different platforms.
- We demonstrate that Ferret-UI 2 advances the UI referring and grounding performance on different platforms. On three categories of tasks (referring, grounding, and user-centric advanced tasks, comprising 9 subtasks $\times$ 5 platforms), Ferret-UI 2 outperforms Ferret-UI, and also shows competitive performance compared to GPT-4o. Besides, Ferret-UI 2 also exhibits strong transfer capabilities across platforms. Finally, Ferret-UI 2 achieved strong performance on recent benchmarks like GUIDE (Chawla et al., 2024) and GUI-World (Chen et al., 2024a).

## 2 RELATED WORK

UI agents have garnered significant attention in recent research, particularly in multimodal models that seek to automate complex UI tasks across diverse platforms. Many works have advanced the field by tackling specific challenges related to single-platform and multi-platform UI understanding, interaction, and automation.

**Single-Platform UI Agents.** Single-platform UI agents focus on automating tasks on a specific device ecosystem, such as Android, iOS, desktop environments, or webpages. On the mobile side, DigiRL (Bai et al., 2024), AppAgent V2 (Li et al., 2024c), AutoDroid (Wen et al., 2024), and MobileFlow (Nong et al., 2024) proposed Android agents targeting on human-like interactions. For web-based agents, systems like WebShop (Yao et al., 2022), WebArena (Zhou et al., 2023), LASER (Ma et al., 2023), WebAgent (Gur et al., 2023), AutoWebGLM (Lai et al., 2024), WebVoyager (He et al., 2024) and Agent-E (Abuelsaad et al., 2024) explored agents that navigate and perform tasks within web environments, while MindSearch agent Chen et al. (2024b) focused on AI engine for web search. AssistGUI (Gao et al., 2023), OS-Copilot (Wu et al., 2024), SYNAPSE (Zheng et al., 2023) and UFO (Zhang et al., 2024a) explored the more complex computer OS interaction. These efforts have significantly improved task-specific automation, although their single-platform nature limits cross-platform flexibility.

**Multi-Platform UI Agents.** Multi-platform UI agents have emerged to address the growing complexity of device ecosystems, supporting a variety of devices and platforms, including mobile, web, and desktop environments. Zheng et al. (2024a); Cheng et al. (2024) demonstrated GPT-4V as a generalist agent when grounded. Recent works like CogAgent (Hong et al., 2023) support UI navi-

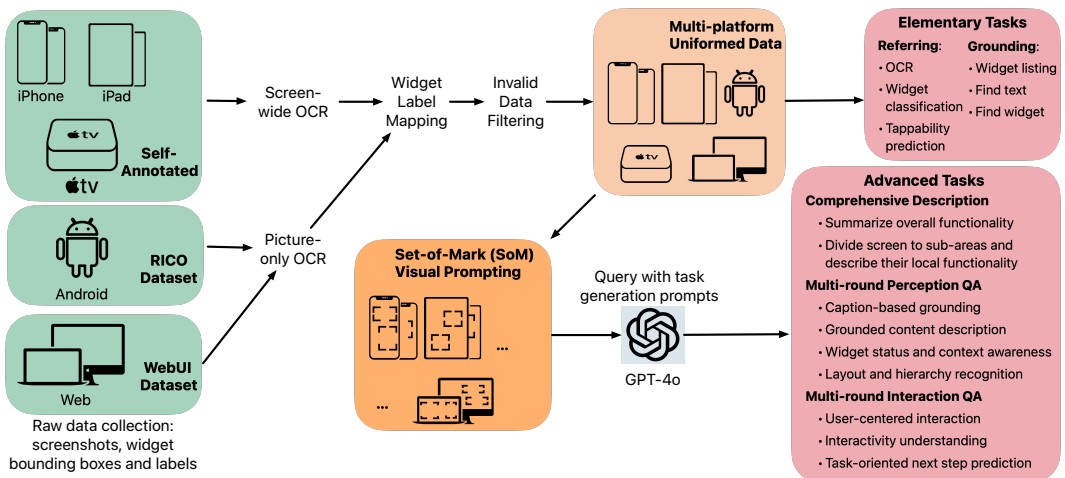

Figure 2: Illustration of the Core-set data generation pipeline.

gation on both PC webpages and Android devices. Mobile-Agent V2 (Wang et al., 2024a) features Harmony OS and Android OS for non-English and English scenarios. Ferret-UI (You et al., 2024) focuses on mobile UI understanding for both Android and iPhone screenshots using multimodal LLMs (McKinzie et al., 2024; Zhang et al., 2024b), with a focus on referring and grounding capabilities. These agents aim to perform more complex, user-intent-based interactions across multiple device types, paving the way for truly generalist multimodal agents.

**UI-Agent Benchmarks.** The evaluation of UI agents requires specialized benchmarks to test various aspects of UI interaction, including task execution, navigation, and interaction understanding. Rico (Deka et al., 2017) remains a foundational dataset for mobile app interaction, while Mobile-Env (Zhang et al., 2023), AndroidEnv (Toyama et al., 2021), AndroidWorld (Rawles et al., 2024a), Android in-the-Wild (Rawles et al., 2024b), AndroidControl (Li et al., 2024b), and AMEX (Chai et al., 2024) provide benchmarks for mobile device control. Windows Agent Arena (Bonatti et al., 2024) introduces a benchmark focusing on PC Windows environment. OSWorld (Xie et al., 2024) takes a broader approach by providing benchmarks for agents in real computer environments, including Ubuntu, MacOS, and Windows. Web-based interaction benchmarks include WebSRC (Chen et al., 2021), Mind2Web (Deng et al., 2024), and WebCanvas (Pan et al., 2024) focusing on structural reading comprehension and task execution in web environments. OmniACT (Kapoor et al., 2024) supports both desktop and web interfaces. More recently, MobileAgentBench (Wang et al., 2024c) and VisualWebBench (Liu et al., 2024b) have introduced taxonomies designed to evaluate the performance of multimodal agents across both mobile and web interfaces. VisualAgentBench (Liu et al., 2024c) expands this with a focus on multimodal LLMs as visual foundation agents. GUI Odyssey (Lu et al., 2024) provides benchmarks for cross-app navigation. GUI-World (Chen et al., 2024a) pioneer in covering multi-platform benchmarking, while CRAB (Xu et al., 2024) further tests cross-environment tasks for GUI agents. These benchmarks contribute to a growing need for unified, multi-platform evaluations that can assess UI agents' adaptability, precision, and efficiency.

Compared to the aforementioned works, Ferret-UI 2 is the first to target universal UI understanding across diverse platforms, including smartphones, tablets, web interfaces, and smart TVs. It focuses on foundational capabilities like fine-grained referring, grounding, and reasoning, aiming to create a generalist agent for versatile UI navigation.

## 3 FERRET-UI 2

In this section, we first describe how we curate our training datasets from the raw data annotations (Section 3.1) and then describe the model architecture (Section 3.2).

### 3.1 DATASET CONSTRUCTION

We construct our own dataset in order to train a strong multi-platform UI understanding model. A flow diagram of our complete dataset generation pipeline is shown in Figure 2.

Table 1: A summary of datasets across various platforms used to train Ferret-UI 2. The screenshot resolution statistics are shown in Appendix F.

| Training Data | Platform | #Images (k) | Task Types |
|---|---|---|---|
| Core-set | iPhone | 112 | Elementary Tasks (Referring, Grounding), Advanced Tasks (Comprehensive Description, Multi-Round Perception QA and Interaction QA) |
| | Android | 61 | |
| | iPad | 19 | |
| | Web | 321 | |
| | AppleTV | 16 | |
| GUIDE | Web | 51 | Next Action Prediction |
| GroundUI-18k | Web | 18 | Simple Interaction |
| Spotlight | Android | 66 | Screen2Word, Widget Caption, Taperception |

**Raw Annotation Collection.** The primary dataset used for training Ferret-UI 2 is a combination of data from various platform types, including iPhone, Android, iPad, Webpage, and AppleTV. The data collection process varies depending on the platform type:

- **iPhone, iPad, and AppleTV:** We use human-collected iPhone, iPad, and AppleTV data under diverse usage scenarios and human-annotated widget bounding box coordinates and labels. To save annotation costs, we do not collect text annotations; instead, text bounding boxes are replaced by screen-wide OCR-detected text and bounding boxes using an OCR confidence threshold of 0.5.

- **Webpage:** The web data is derived from the WebUI dataset (Wu et al., 2023). Bounding boxes of all types of UI widgets and text annotations for non-picture widgets are directly parsed from the source HTML view hierarchy tree, providing high-quality annotations. For picture widgets, we further use OCR to detect texts contained in the pictures.

- **Android:** The Android data for screenshots, bounding boxes, and text annotations is transformed from the RICO dataset (Deka et al., 2017). Similar to the WebUI dataset, we also perform picture-only OCR to complete the missing text annotations in picture widgets.

Use of this data for the purpose of this paper was approved by our organization's legal team.

For all the collected data, we perform data filtering including: ($i$) filter out or narrow down out-of-bound bounding boxes and remove empty screenshots with no remaining bounding boxes after box filtering; ($ii$) since we do not intend to add multilingual support for the Ferret-UI 2 model, screenshots with more than $5\%$ non-ASCII characters in the text annotations are also removed.

Despite the different types of label spaces from various sources, we filter out bounding boxes associated with less relevant labels (*e.g.*, UI types) and uniformly map the remaining labels to a common label space containing 13 classes: 'Checkbox', 'Button', 'Container', 'Dialog', 'Icon', 'PageControl', 'Picture', 'SegmentedControl', 'Slider', 'TabBar', 'Text', 'TextField', and 'Toggle', and obtain a multi-platform uniformed dataset with raw UI widget annotations. We provide the original label statistics and their converted labels in Appendix G.

We name the above-collected screenshot dataset the Core-set, which we will use to construct the elementary and advanced task data. Besides, we also employ third-party training datasets to enrich our data source and avoid overfitting our predefined tasks. A complete statistics of the training dataset of Ferret-UI 2 is summarized in Table 1, indicating that the dataset distribution is very unbalanced across different platforms. In particular, the number of iPad and AppleTV screenshots is significantly smaller than that of other platforms. To address this, we ($i$) assign different loss weights to different platforms during training, and ($ii$) generate all three types of advanced tasks for each example of iPad and AppleTV platforms and generate only 1 type of advanced task for each example of other platforms.

Compared to the training dataset of Ferret-UI, which relies on model-detected bounding boxes, Ferret-UI 2's training dataset predominantly utilizes either human-collected annotations or bounding boxes directly parsed from the source HTML, resulting in a significant improvement in annotation quality, evidenced later in our quantitative evaluation in Section 4.2.

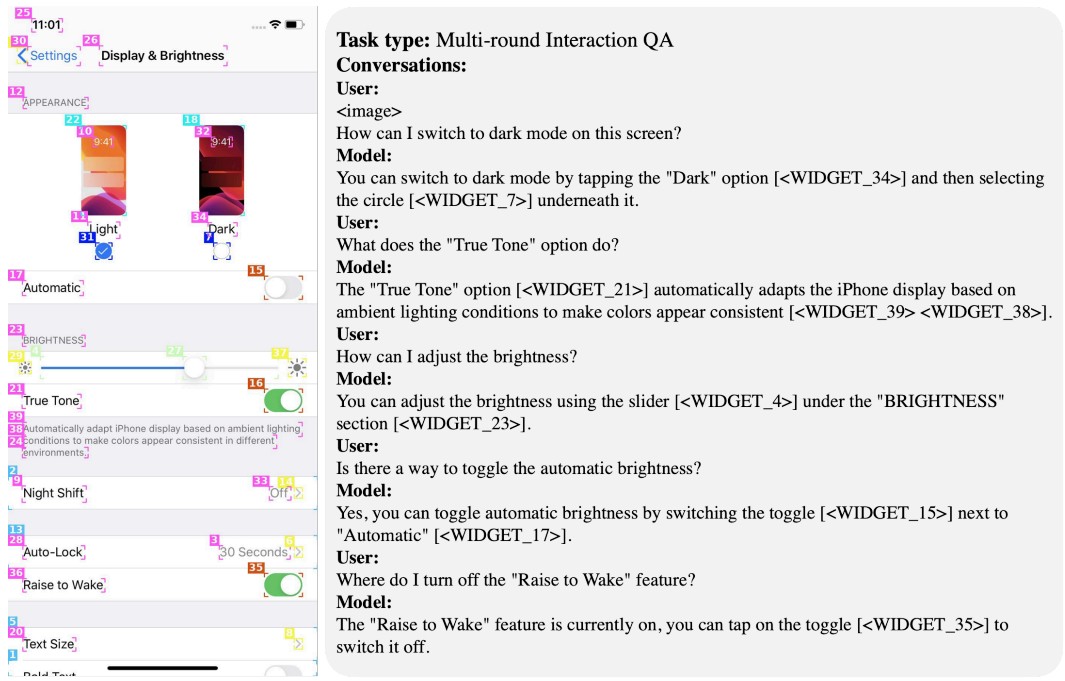

Figure 3: Example of set-of-mark visual prompting (Yang et al., 2023) (**left**) and one of its generated advanced task training examples (**right**).

**Task Data Generation.** For task data generation, we follow the paradigm of Ferret-UI data construction, which includes both elementary tasks and advanced tasks.

Elementary tasks (Figure 2) consist of 3 referring tasks and 3 grounding tasks. Specifically, referring tasks include ($i$) *OCR*: recognizing the text given a text bounding box, ($ii$) *widget classification*: predict the UI type of the elements, and ($iii$) *tapperbility*: predict whether the selected widget is tappable for interaction; meanwhile, grounding tasks include ($i$) *widget listing*: list all the widgets in the screen, ($ii$) *find text*: find the location of a given text, and ($iii$) *find widget*: find the widget given the widget description.

For advanced tasks, we prompt GPT-4o with bounding box annotations of a given screenshot and require GPT-4o to generate QA tasks related to the UI widgets in the screenshot. Unlike Ferret-UI, which mainly focuses on spatial descriptions due to the limitation of using textual prompts without image information (*i.e.*, screenshots) for bounding box annotations, Ferret-UI 2 leverages GPT-4o to generate advanced task data that covers a variety of aspects of UI understanding. This is possible because GPT-4o demonstrates an improved ability to comprehend the spatial relationships between UI widgets when provided with the screenshot as input. Specifically, we prompt GPT-4o to generate 3 types of advanced tasks (shown in Figure 2) including: ($i$) *comprehensive description*: describe global and local functionalities of the screen, ($ii$) *multi-round perception QA*: multi-round question answering regarding the UI perception capabilities, and ($iii$) *multi-round interaction QA*: multi-round question answering regarding the single-step and user-centric UI interactions based on the current screen status. More detailed requirements and prompts for GPT-4o when generating advanced tasks are provided in Appendix C.

We empirically find it hard for GPT-4o to find the location of referred UI widgets with the original screenshot as input (*i.e.*, bad grounding capability). To address this, we use Set-of-Mark (SoM) visual prompting (Yang et al., 2023) when generating multi-round perception and interaction QA training samples. One example of SoM prompting and its generated data sample is shown in Figure 3, where each UI widget is marked with a corner-style bounding box and a unique number tag for easy identification. Furthermore, the same class of UI widgets have the same color for visual prompting to help GPT-4o better differentiate the bounding boxes of spatially close or nested widgets. Please refer to Figure 5 for visual prompts on other platforms, and Appendix D for additional examples of generated data for advanced tasks.

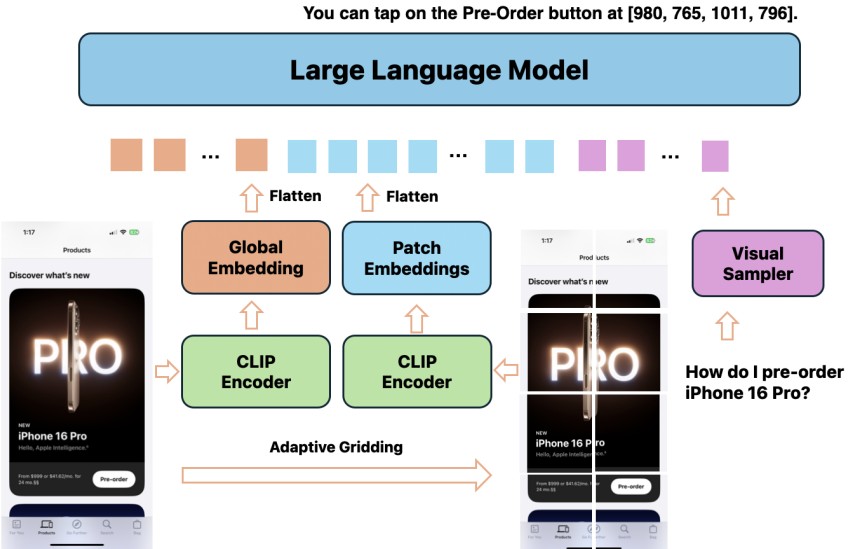

Figure 4: Overview of the Ferret-UI 2 model architecture, which allows for seamless UI understanding and user-centered single-step interactions with high-resolution support.

---

**Algorithm 1:** Adaptive $N$-gridding

---

**Require:** Original resolution: w × h, grid size: $336 \times 336$, size limit $N$
**Ensure:** Optimal gridding size $N_w$ and $N_h$ $(N_w, N_h \in \mathbb{N}^+)$
1: $N_{w_{\text{best}}}, N_{h_{\text{best}}} \leftarrow 0, \Delta_{\text{best}} \leftarrow \infty, N_{w_0} \leftarrow \frac{w}{336}, N_{h_0} \leftarrow \frac{h}{336}$
2: **for** $N_w = 1$ to $N$ **do**                                             ▷ Traverse all grid configuarations
3:     **for** $N_h = 1$ to $N - N_w$ **do**
4:         $\Delta_{\text{aspect}} \leftarrow \sqrt{\left|\frac{N_w}{N_h} - \frac{N_{w_0}}{N_{h_0}}\right| \left|\frac{N_h}{N_w} - \frac{N_{h_0}}{N_{w_0}}\right|}$          ▷ Get aspect ratio change
5:         $\Delta_{\text{pixel}} \leftarrow \frac{\left|N_w \times N_h - N_{w_0} \times N_{h_0}\right|}{N_{w_0} \times N_{h_0}}$          ▷ Get relative pixel change for resizing
6:         **if** $\Delta_{\text{best}} > \Delta_{\text{aspect}} \times \Delta_{\text{pixel}}$ **then**
7:             $(N_{w_{\text{best}}}, N_{h_{\text{best}}}) \leftarrow (N_w, N_h)$
8:             $\Delta_{\text{best}} \leftarrow \Delta_{\text{aspect}} \times \Delta_{\text{pixel}}$
9:         **end if**
10:     **end for**
11: **end for**
12: **return** $(N_{w_{\text{best}}}, N_{h_{\text{best}}})$

---

As aforementioned in Table 1, in addition to the tasks generated on the Core-set, we augment training data with additional third-party datasets, including GroundUI-18k (Zheng et al., 2024b), GUIDE (Chawla et al., 2024) and Spotlight (Li & Li, 2023).

## 3.2 MODEL ARCHITECTURE

As shown in Figure 4, the model architecture of Ferret-UI 2 directly builds upon Ferret-UI (You et al., 2024), which uses the Any-Resolution (AnyRes) method (Liu et al., 2024a) to enhance referring and grounding, enabling the encoder to capture diverse image resolutions.

Specifically, the CLIP image encoder first extracts both global (derived from the low-resolution overview image) and local features (corresponding to high-resolution sub-images) from the UI screenshot. Then, these image features are flattened and sent into the LLM. The Visual Sampler identifies and selects the relevant UI regions based on user instructions. The model then outputs grounded descriptions for perception or interaction with the UI elements.

**Adaptive Gridding.** Local image features are extracted by calculating the optimal grid size using our proposed *adaptive N-gridding* mechanism, then resizing and encoding the visual features of

Table 2: Results on our constructed benchmarks for elementary and advanced tasks, as well as the GUIDE benchmark (Chawla et al., 2024). Results on elementary and advanced tasks are averaged over all platforms, including iPhone, Android, iPad, Webpage, and AppleTV. Each platform includes 6 elementary tasks and 3 advanced tasks. SeeClick model (Cheng et al., 2024) trained on their original data is compared. (†) In tasks that require referring, GPT-4o is equipped with set-of-mark (SoM) prompting by adding a red rectangular box to screenshots for the referred widget. Note that SoM visual prompting is not used for Ferret-UI and Ferret-UI 2.

| Model | Backbone | Elementary | | Advanced | | GUIDE Bench | |
|---|---|---|---|---|---|---|---|
| | | Refer | Ground | GPT-4o Score | Multi-IoU | BertScore | IoU |
| Ferret-UI | Vicuna-13B | 64.15 | 57.22 | 45.81 | 18.75 | 41.15 | 26.91 |
| Ferret-UI 2 | Gemma-2B | 75.20 | 78.13 | 80.25 | 40.51 | 83.71 | 51.13 |
| | Llama3-8B | 80.28 | **82.79** | **89.73** | 41.15 | **91.37** | **55.78** |
| | Vicuna-13B | **81.34** | 81.31 | 86.25 | **41.71** | 88.81 | 54.71 |
| SeeClick (Cheng et al., 2024) | QWen-VL-9.6B | 51.58 | 62.82 | 67.49 | 21.56 | 54.70 | 39.51 |
| GPT-4o | - | 56.47 | 12.14 | 77.73 | 7.06 | 75.31 | 9.64 |
| GPT-4o + SoM-Prompt† | - | **87.91** | - | 84.33 | 7.36 | - | - |

each grid. This is a key model innovation compared to Ferret-UI. As shown in Algorithm 1, the optimal gridding size $N_w$ and $N_h$ is determined when the gridding and resizing based on $(N_w, N_h)$ lead to minimal *aspect ratio change* times *the relative pixel number change*, under the constraint $N_w + N_h \leq N$, where $N$ is the *size limit*. With the size limit $N$, the total number of grids is upper bounded by $\lfloor \frac{N^2}{4} \rfloor$. Compared to the AnyRes module that has unbounded cost, the key differences of adaptive $N$-gridding is it automatically finds the optimal gridding configuration, *i.e.*, least resolution distortion (aspect ratio change and pixel number change) within a predefined inference cost limit that $N_w \times N_h \leq \lfloor \frac{N^2}{4} \rfloor$, which is both information-preserving and efficient for local encoding. With adaptive gridding, Ferret-UI 2 understands UI screens and provides user-centered interactions with an optimal configuration at any resolution given the inference cost limit specified as $N$.

## 4 EXPERIMENTS

### 4.1 EXPERIMENT SETUP

**Training Data.** The training datasets are summarized in Table 1, which can be divided into two categories: ($i$) datasets constructed by our own, which include elementary task data and advanced task data across all platforms as introduced in Section 3.1, and ($ii$) public datasets including GroundUI-18k (Zheng et al., 2024b), a simple user-centered interaction dataset on webpage screenshots, GUIDE (Chawla et al., 2024), a next-action prediction dataset on webpage screenshots, and Spotlight (Li & Li, 2023), an Android UI understanding and interaction dataset.

**Model.** Following Ferret-UI (You et al., 2024), Ferret-UI 2 uses a CLIP ViT-L/14 model as the image encoder; for the LLM backbone, besides Vicuna-13B (Chiang et al., 2023) as used in the original Ferret-UI, we also tried 2 additional LLMs at mobile scales, including Gemma-2B (Team et al., 2024) and Llama3-8B (Dubey et al., 2024). As to dynamic high-resolution image encoding, we set the size limit $N$ to 8, so that the maximal grid number is 16 for adaptive gridding.

**Evaluation.** At a high level, model evaluation falls into two broad categories: ($i$) benchmarks we constructed, and ($ii$) public benchmarks. For our benchmarks, we created a total of 45, including 6 elementary tasks and 3 advanced tasks per platform type, across 5 platforms. For elementary tasks, we follow the evaluation metrics outlined by You et al. (2023). For advanced tasks, we use GPT-4o to score generated answers for a given screenshot and user query, visually prompting GPT-4o with a red rectangular bounding box. Advanced tasks are tested using GPT-4o evaluation score and multi-IoU. The multi-IoU is calculated by first matching predicted bounding boxes with ground truth bounding boxes and then calculating the average IoU of each pair of bounding boxes (IoU = 0 if no match). Furthermore, we conduct a next-action prediction test given previous action history on the GUIDE benchmark (Chawla et al., 2024), and evaluate the semantic similarity w.r.t. the reference answer and grounding Intersection-over-Union (IoU). Additionally, we evaluate our model on the recently released GUI-World benchmark (Chen et al., 2024a) on the supported platforms following

Table 3: Zero-shot performance of Ferret-UI 2 on the GUI-World benchmark (Chen et al., 2024a).

| Model | GPT-4 Score | | | |
|---|---|---|---|---|
| | iOS | Android | Webpage | Average |
| MiniGPT4Video (Ataallah et al., 2024) | 1.501 | 1.342 | 1.521 | 1.455 |
| VideoChat2 (Li et al., 2024a) | 2.169 | 2.119 | 2.221 | 2.170 |
| Chat-Univi (Jin et al., 2024) | 2.337 | 2.390 | 2.349 | 2.359 |
| GUI-Vid (Chen et al., 2024a) | 2.773 | 2.572 | 2.957 | 2.767 |
| QWen-VL-MAX (Bai et al., 2023) | 2.779 | 2.309 | 2.656 | 2.580 |
| SeeClick Cheng et al. (2024) | 2.614 | 2.650 | 2.848 | 2.704 |
| Ferret-UI (You et al., 2024) | 2.713 | 2.791 | 2.411 | 2.638 |
| Ferret-UI 2 | **2.881** | **2.954** | **3.013** | **2.948** |
| Gemini-Pro 1.5 (Reid et al., 2024) | 3.213 | 3.220 | 3.452 | 3.295 |
| GPT-4o | 3.558 | 3.561 | 3.740 | 3.619 |

Table 4: Zero-shot cross-platform transfer results of Ferret-UI 2. For simplicity, we train and test the model only using data corresponding to the elementary tasks.

| Training | Test - Referring | | | | | Test - Grounding | | | | |
|---|---|---|---|---|---|---|---|---|---|---|
| | iPhone | iPad | AppleTV | Web | Android | iPhone | iPad | AppleTV | Web | Android |
| iPhone | **86.3** | 68.1 | 31.2 | 45.3 | 71.2 | **84.1** | 65.2 | 43.1 | 51.7 | 63.1 |
| iPad | 67.5 | **80.2** | 40.7 | 51.5 | 63.3 | 64.5 | **82.1** | 32.1 | 38.5 | 53.8 |
| AppleTV | 29.1 | 45.1 | **79.3** | 54.2 | 36.4 | 33.7 | 41.2 | **81.6** | 52.1 | 29.7 |
| Web | 59.2 | 57.4 | 41.2 | **85.5** | 41.7 | 54.0 | 51.2 | 46.5 | **87.5** | 45.9 |
| Android | 72.5 | 60.7 | 35.7 | 51.2 | **86.2** | 66.7 | 48.9 | 29.7 | 44.1 | **83.9** |

the original GPT-4 evaluation protocol as in Chen et al. (2024a), which does not include evaluating the grounding capability for interaction-related UI tasks.

## 4.2 EXPERIMENT RESULTS

**Main results.** Our main results are summarized in Table 2, which shows the comparative performance of different models on our constructed elementary and advanced tasks, as well as the GUIDE benchmark (Chawla et al., 2024). Note, that for each data entry corresponding to the elementary and advanced tasks, it is an average across all platforms. The detailed results on each platform are provided in Table 6 of Appendix A. Below, we highlight a few observations.

- Ferret-UI 2, powered by Llama-3-8B, delivers the best results across most metrics. It achieves the highest GPT-4o score on advanced tasks, with a notable 89.73, surpassing Ferret-UI by 43.92 points and GPT-4o by 12.0 points. Notably, Ferret-UI 2 with Llama-3-8B also achieves the highest IoU score on the GUIDE benchmark with 55.78, indicating superior grounding capability.

- Ferret-UI 2, equipped with Vicuna-13B, also performs well, *e.g.*, achieving a strong Multi-IoU score of 41.71 on advanced tasks. Despite being six times smaller, Ferret-UI 2 with Gemma-2B delivers competitive performance across the board.

- In contrast, GPT-4o struggles with fine-grained UI understanding, as shown by its low referring (56.47) and grounding (12.14) scores in the elementary tasks. Its Multi-IoU and IoU scores in advanced tasks and the GUIDE benchmark are also low.

Overall, the results demonstrate the versatility of Ferret-UI 2 in handling UI understanding tasks across different platforms.

**Results on GUI-World.** To further demonstrate the zero-shot performance of using Ferret-UI 2 out of the box, we further test our model on the recently released GUI-World benchmark (Chen et al., 2024a). Results are summarized in Table 3. Clearly, Ferret-UI 2 does not overfit the training data and can generalize well to the test data in the wild. Notably, Ferret-UI 2 outperforms GUI-Vid (Chen et al., 2024a), a model developed in the GUI-World paper, on supported platforms including iOS, Android, and Webpages.

Table 5: Ablation results of the architecture and dataset improvements of Ferret-UI 2 w.r.t. Ferret-UI-anyRes (You et al., 2024), *i.e.*, the high-resolution version of Ferret-UI equipped with the any-resolution module. **iPhone v1** refers to the dataset on the iPhone platform originally used by Ferret-UI, while **iPhone v2** is the data used by Ferret-UI 2. Models are evaluated on the advanced tasks.

| Training Data | Model | iPhone v1 | | iPhone v2 | |
|---|---|---|---|---|---|
| | | GPT-4o Score | Multi-IoU | GPT-4o Score | Multi-IoU |
| iPhone v1 | Ferret-UI-anyRes | 91.3 | 36.89 | 68.3 | 27.13 |
| | Ferret-UI 2 | 93.7 (+2.4) | 37.12 (+0.23) | 70.2 (+1.9) | 28.21 (+1.08) |
| iPhone v2 | Ferret-UI-anyRes | 86.2 | 35.89 | 85.97 | 39.81 |
| | Ferret-UI 2 | 88.1 (+1.9) | 36.43 (+0.54) | 89.7 (+3.73) | 41.73 (+1.92) |

## 4.3 Ablation Study

**Cross-Platform Transferability.** In Table 4, we evaluate the zero-shot platform (domain) transfer capability of the Ferret-UI 2 model by training it on elementary tasks from one platform and testing it on other platforms. These results provide insights into how well the model generalizes across different platforms. We observe similar performance patterns across tasks. Specifically,

- iPhone transfers well to iPad and Android platforms on both tasks, achieving at least 68.1 and 71.2 scores on referring tasks and 65.2 and 63.1 scores on grounding tasks due to its diverse screenshot contents (because of the large screenshot number) and similar resolutions and aspect ratios with other two platforms. iPad and Android also transfer fairly well to iPhone at around 65 scores.

- AppleTV and Web do not transfer very well to the mobile domains, including iPhone, Android, and iPad, achieving the highest referring score of 59.2 and grounding score of 54.0, possibly because they are mostly landscape screenshots, which is in contrast to the mostly portrait screenshots in mobile platforms. Models trained on other domains all achieve poor performance on the AppleTV test domain, with the highest score of around 40 on both kinds of tasks, which is reasonable due to a large domain gap in terms of AppleTV's content distribution compared to other domains.

The results suggest that ($i$) iPhone, iPad and Android have similar content distribution, which helps them generalize to each other; ($ii$) models trained on more diverse contents (*e.g.*, around 100k iPhone data) can generalize better to other platforms; ($iii$) platforms with similar resolution and aspect ratios may transfer better to each other; and ($iv$) good transferability among some of these platforms contributes to the cross-platform performance of Ferret-UI 2.

**Ablation on Architecture and Dataset Improvements.** Table 5 presents a comparison between the Ferret-UI and Ferret-UI 2 model trained on different versions of the iPhone dataset. Models are evaluated on the test set corresponding to advanced tasks. The results indicate that both the adaptive $N$-gridding and improved dataset (iPhone v2) contribute to performance gains.

Specifically, when evaluated on the iPhone v1 test set, Ferret-UI 2 shows a slight improvement over Ferret-UI, with the GPT-4o score increased from 91.3 to 93.7 and the Multi-IoU score increased from 36.89 to 37.12. However, the improvements are more pronounced on the iPhone v2 dataset. Here, Ferret-UI 2 achieves a GPT-4o score of 89.7, outperforming Ferret-UI's 85.97, along with a substantial Multi-IoU score boost from 39.81 to 41.73. These results suggest that while both architecture and dataset enhancements contribute to overall performance, the new dataset plays a more significant role in driving improvements, particularly on more challenging tasks.

## 5 Conclusions

In this paper, we presented Ferret-UI 2, a multimodal large language model designed to improve UI understanding and interaction across diverse platforms. With multi-platform support, high-resolution image encoding with adaptive gridding, and improved data generation, Ferret-UI 2 outperforms Ferret-UI on all tested benchmarks. The model demonstrates strong zero-shot transferability across platforms, establishing Ferret-UI 2 as a solid foundation for universal UI understanding. Future work will focus on incorporating additional platform types and building a generalist agent for universal UI navigation.

## ETHICS STATEMENT

The use of UI understanding data for this paper was approved by our organization's legal team. We have strictly adhered to ethical standards, ensuring that no private or sensitive information has been used or compromised.

## ACKNOWLEDGEMENTS

The authors would like to thank Forrest Huang, Zhen Yang, Haoxuan You, Amanda Swearngin, Alexander Toshev and Yang Zhao for valuable guidance, suggestions, and feedback.

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

## A  DETAILED RESULTS ON ELEMENTARY AND ADVANCED TASKS

Table 6 shows the performance on 9 subtasks × 5 platforms of the Ferret-UI 2 model. We empirically find that OCR, widget listing, and comprehensive description are relatively difficult tasks compared to other tasks within the same task type.

Table 6: The performance breakdown of Ferret-UI 2 with Llama-3-8B backbone. Note that we only report GPT-4o evaluation scores and omit the multi-IoU scores for advanced tasks for simplicity.

| Task type | Task | Test Domain | | | | |
|---|---|---|---|---|---|---|
| | | iPhone | iPad | AppleTV | Web | Android |
| | OCR | 75.3 | 69.3 | 74.3 | 80.5 | 74.5 |
| Refer | Widget Classify | 79.1 | 78.7 | 82.5 | 83.6 | 82.0 |
| | Tapperbility | 89.2 | 86.0 | - | 84.3 | 85.6 |
| | Widget Listing | 76.7 | 74.9 | 71.6 | 79.7 | 76.8 |
| Ground | Find Text | 85.2 | 84.0 | 82.2 | 90.2 | 82.2 |
| | Find Widget | 88.6 | 86.8 | 87.7 | 87.8 | 86.4 |
| | Comprehensive | 89.0 | 86.5 | 84.1 | 83.7 | 85.6 |
| Advanced Tasks | Perception | 94.1 | 87.5 | 86.6 | 93.2 | 92.1 |
| | Interaction | 93.5 | 92.7 | 86.1 | 96.4 | 94.7 |

## B  GROUNDING PERFORMANCE ON SCREENSPOT BENCHMARK

We evaluate Ferret-UI 2 on the ScreenSpot (Cheng et al., 2024) benchmark for grounding performance on unseen data. Results are shown in Table 7. Ferret-UI 2 achieves 54.0% average accuracy, outperforming CogAgent (Hong et al., 2023) and SeeClick (Cheng et al., 2024), and in particular, achieving good performance on our supported Mobile and Web platforms while achieving fair performance on the unseen Desktop platform.

Table 7: Performance of Ferret-UI 2 on the ScreenSpot benchmark (Cheng et al., 2024). **I/W** refers to Icon/Widget data type. Note that the data distribution of ScreenSpot is unseen during Ferret-UI 2 training. The results indicate that Ferret-UI 2 achieve superior grounding accuracy on Mobile and Web screenshots, while underperforms other two models on the unseen Desktop screenshots, which means Ferret-UI 2 transfers well to unseen data in supported platforms.

| Model | Model Size | Mobile | | Desktop | | Web | | Avg |
|---|---|---|---|---|---|---|---|---|
| | | Text | I/W | Text | I/W | Text | I/W | |
| CogAgent | 8B | 67.0% | 24.0% | **74.2%** | 20.0% | 70.4% | 28.6% | 47.4% |
| SeeClick | 9.6B | 78.0% | 52.0% | 72.2% | **30.0%** | 55.7% | 32.5% | 53.4% |
| Ferret-UI 2 | 8B | **80.3%** | **55.4%** | 52.1% | 21.7% | **81.2%** | **33.5%** | **54.0%** |

## C  DETAILED GPT-4O PROMPTS FOR GENERATING ADVANCED TASK DATA

In this section, we elaborate on how we prompt GPT-4o to generate training data for advanced tasks. In particular, we have the following requirements when generating each type of advanced task.

1. **Comprehensive Description**: Provide a one-sentence description of the overall functionality of the UI page shown in the screenshot. Then, describe the screenshot in detail by dividing it into several areas/groups and explaining the functionality of each area/group.

2. **Multi-Round Perception QA**:

Table 8: Statictics of different multi-platform GUI understanding datasets.

| Dataset | # Sample | Platforms | Tasks |
|---|---|---|---|
| OmniAct | 9,802 | Desktop, Web | Code Generation |
| OS-World | 369 | Desktop, Web | General Control |
| AITW | 715,142 | Android(Apps+Web) | Navigation |
| GUI-world | 12,379 | iPhone, Android, Web, XR, Desktop | GUI understanding, instruction following |
| Ferret-UI | 123,702 | iPhone, Android | Elementary: Referring, Grounding Advanced: Function/detailed description, perception and interaction QA |
| Ferret-UI 2 | 529,411 | iPhone, Android, iPad, AppleTV, Web | Elementary: Referring, Grounding Advanced: Comprehensive description, perception and interaction QA |

(a) **Basic Perceptual Understanding**: Interpret the content of referred widgets, ground the positions of the widgets based on the widget descriptions, and describe the functionality and status (enabled, disabled, selected, hovered) of each widget.

(b) **Contextual Awareness**: Understand the context in which UI widgets are presented, including the relationships between different UI components and how they contribute to the overall user experience.

(c) **Layout and Hierarchy Recognition**: Recognize how widgets are grouped and nested within the UI layout.

3. **Multi-Round Interaction QA**:

(a) Generate instructions for user-centered interaction. For example, "Please help me confirm submission" instead of "please click on [Box0] button".

(b) Identify and interact with buttons, links, icons, scrollbars, and toggles, and recognize widgets that cannot be interacted with, such as static text and background images.

(c) Identify and interact with input fields, dropdowns, checkboxes, and radio buttons, as well as their current states (*e.g.*, checked, unchecked, filled, empty) in the context of specific interaction tasks.

(d) Predict the current step for potential navigation paths based on currently visible widgets like tabs, menus, and breadcrumbs controls.

# D ADVANCED TASK EXAMPLES



**Comprehensive Description**

**OVERALL**: This UI page is for adjusting settings related to downloads, language, news, and notifications.
**DETAIL:**

• **Top Navigation Bar:** Contains menu options such as Home, Account, Settings (currently selected), and Downloads, allowing users to navigate between different sections.

• **Games Section:** Features a checkbox labeled "Download over WiFi only," which is currently checked, indicating that downloads are restricted to WiFi connections.

• **Language Section:** Offers a dropdown menu to select the language, currently set to English, with a pink arrow indicating more options.

• **Title Screen Section:** Includes a checkbox for "Show Sago News," which is checked, meaning news updates will be displayed on the title screen.

• **Notifications Section:** Contains a checkbox for "Display notifications," which is unchecked, allowing the user to enable or disable notifications.

• **Close Button:** Located in the top right corner, represented by an 'X', used to close the settings page.



---

**Multi-Round Perception QA**

**Q:** *What does the "SERVEWARE" text represent?*
**A:** The text "SERVEWARE" [718, 72, 948, 108] is likely a category label indicating that the items displayed are related to serving ware.

**Q:** *Can you identify any product names and their corresponding prices?*
**A:** Yes, the product "ARV BRÖLLOP" [35, 847, 233, 907] is priced at $12.99 [10, 6, 112, 32] and "IKEA 365+" [609, 787, 809, 823] is priced at $2.99 [1174, 1336, 1627, 2119].

**Q:** *What is the function of the shopping cart icon?*
**A:** The shopping cart icon [630, 2157, 690, 2217] is typically used to view the items in your cart or proceed to checkout.

**Q:** *How is the product rating displayed?*
**A:** The product rating is displayed with star symbols and a numerical rating. For example, "ARV BRÖLLOP" has a rating of 4.9 next to the stars [[38, 956, 100, 986], [248, 1004, 294, 1030]].

**Q:** *What does the "Filter" button do?*
**A:** The "Filter" button [1564, 170, 1630, 198] is likely used to apply specific criteria to narrow down the product search results.

**Q:** *Are there any navigation elements on the screen?*
**A:** Yes, there seems to be a navigation bar at the bottom with icons for different sections, like a house icon for home [611, 268, 1057, 714] and a pencil icon [1320, 2157, 1380, 2217] which might be for editing or creating a list.

---

**Multi-Round Interaction QA**

**Q:** *How can I refresh the content shown?*
**A:** You can tap on the refresh icon [272, 736, 424, 882] to reload the content.

**Q:** *What does the "General" tag refer to?*
**A:** The "General" tag [1074, 900, 1443, 926] categorizes the content related to general events, like the Caltech Beaver Name Reveal.

**Q:** *How can I view more recent activities?*
**A:** You can tap the "Recent" section label [90, 598, 202, 630] to view more recent activities.

**Q:** *What is the logo displayed on this page?*
**A:** The logo at the top of the page is the SCIAC Network logo [198, 56, 444, 128] representing the network conducting the event.

**Q:** *How do I check the institution hosting the event?*
**A:** The event is hosted by the California Institute of Technology, as shown in the text [11, 499, 483, 542] below the event details.

## E  DATASET COMPARISON

In this section, we compare the dataset of Ferret-UI 2 with previous multi-platform datasets as shown in Table 8. In particular, we outline the data generation pipeline of Ferret-UI (You et al., 2024) in Figure 6. Through a comparison of Figure 2 and 6, we highlight the following key differences between Ferret-UI 2 and Ferret-UI:

- **Multi-platform support**: The training data of Ferret-UI 2 consists of 5 platforms compared to 2 platforms of Ferret-UI.

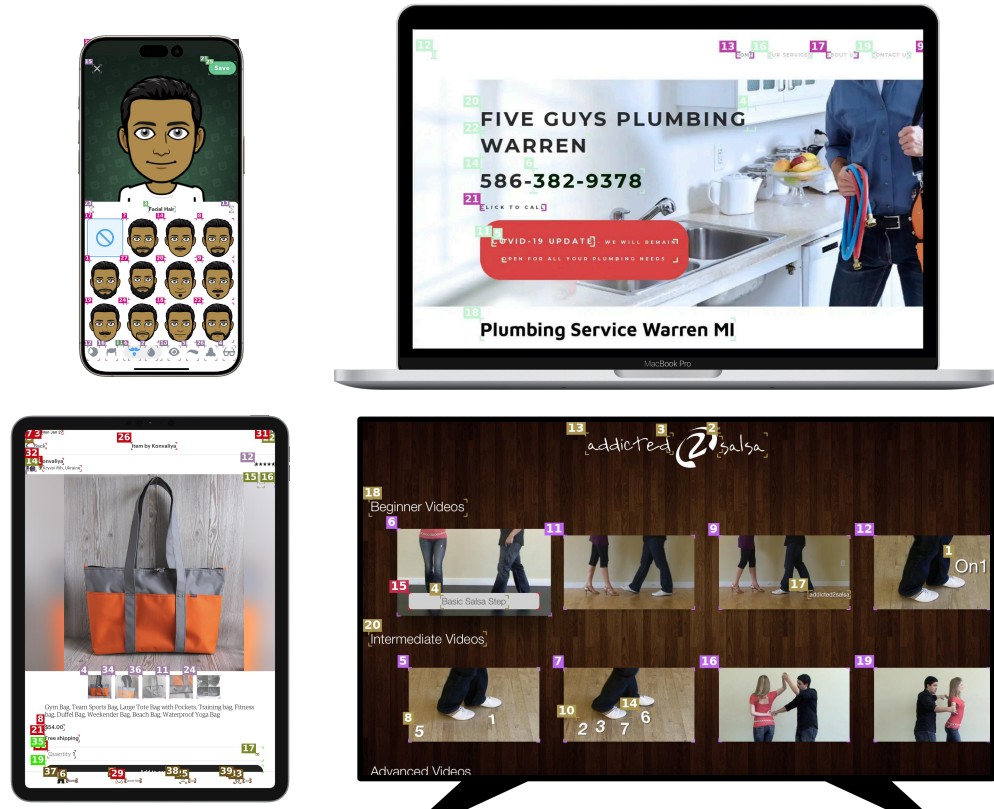

Figure 5: Examples of visual prompting using GPT-4o to generate task data for Multi-Round Perception QA and Multi-Round Interaction QA. Each UI widget is annotated with a corner-style bounding box, where only the corners of the widget are highlighted by small lines, leaving the rest of the box open. This minimalistic bounding style is accompanied by a unique number tag placed near one of the corners, making it easy to identify and reference specific UI widgets for further interaction or perception analysis.

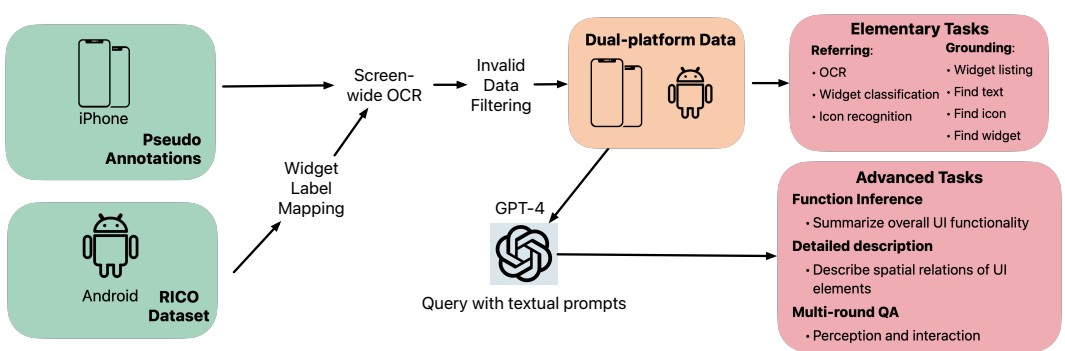

Figure 6: Illustration of the data generation pipeline of Ferret-UI (You et al., 2024).

- **Raw annotation qualities**: The majority part of bounding boxes, labels, and on-screen text annotations of Ferret-UI 2 are either extracted from source data or annotated by humans, while these annotations of Ferret-UI are all generated from model detection.

- **Bounding box prompting**: When generating advanced tasks, Ferret-UI 2 uses GPT-4o+SoM visual prompting for bounding boxes, while Ferret-UI uses purely textual prompting containing coordinates, bbox labels, and text annotations.

- **Advanced task quality**: Due to the constraint of textual prompting, the Ferret-UI pipeline cannot perceive the visual content of UI elements, thus limiting the quality and content diversity of generated advanced tasks.

## F  RESOLUTION STATISTICS

In this section, we present the resolution statistics for images collected from various platforms, categorized by device types and resolutions, as summarized in Table 9.

Table 9: Resolution statistics by device type.

| Device Type | Resolution | Number of Images |
|---|---|---|
| iPhone | 828×1792 | 83,250 |
| | 1125×2436 | 6,055 |
| | 1792×828 | 4,686 |
| | 2436×1125 | 104 |
| iPad | 2224×1668 | 4,829 |
| | 1668×2224 | 14,312 |
| | 1242×2208 | 19 |
| AppleTV | 1920×1080 | 16,152 |
| WebUI | 1280×720 | 53,500 |
| | 1366×768 | 53,500 |
| | 1536×864 | 53,500 |
| | 1920×1080 | 53,500 |
| | 2048×2732 | 53,500 |
| | 1170×2532 | 53,500 |
| Android | 540×960 | 14,092 |
| | 1080×1920 | 52,102 |
| | 1920×1080 | 55 |
| | 960×540 | 12 |

## G  LABEL STATISTICS AND MAPPING RESULTS OF ORIGINAL DATA ACROSS PLATFORMS

We demonstrate the label statistics of original data across platforms and their mapping results into a uniform label space for better joint training as shown in Figures 7, 8, 9 and 10. The mapping results are obtained via GPT-4 suggestions and additional human reviews. Note that the "Other" label after mapping indicates the widget information will be deprecated.

## H  ADDITIONAL INFERENCE EXAMPLES

In Figure 11 to 14, we show additional qualitative inference examples of Ferret-UI 2 on different platforms. Additionally, in Figure 15, we show an example of Ferret-UI 2 performing multi-step interactions on real-time webpages following GUIDE-style QAs.

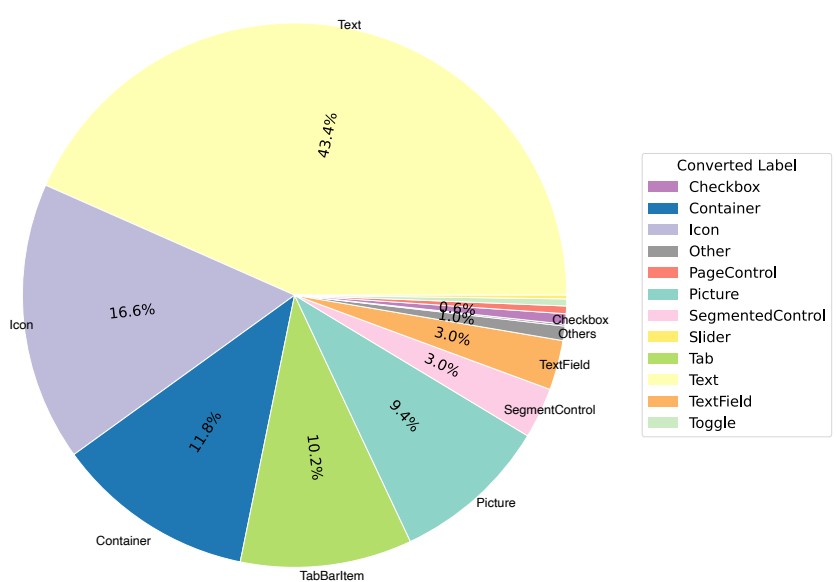

Figure 7: The statistics of original labels (**wedge**) and converted labels (**legend**) on iOS (iPhone + iPad) platforms. Each color represents one converted label, and each wedge represents one original label from source data.

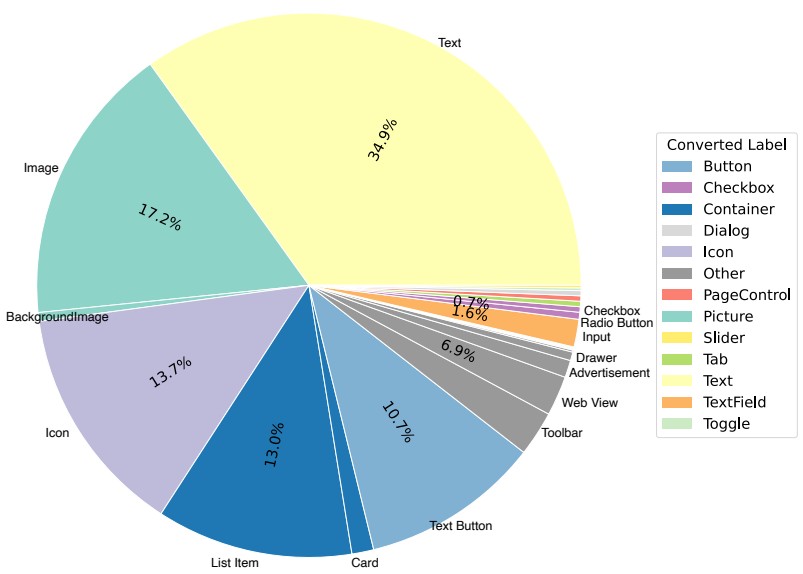

Figure 8: The statistics of original labels (**wedge**) and converted labels (**legend**) on Android platform. Each color represents one converted label, and each wedge represents one original label from source data.

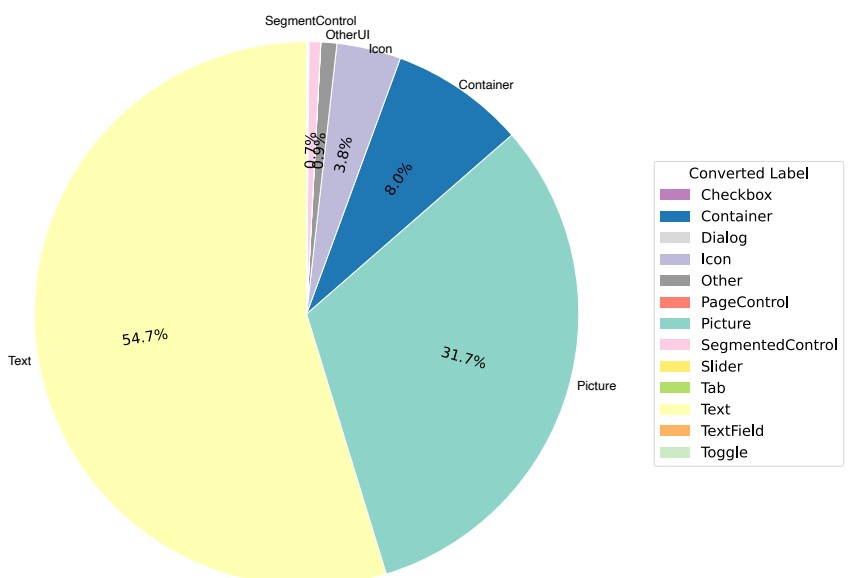

Figure 9: The statistics of original labels (**wedge**) and converted labels (**legend**) on AppleTV platform. Each color represents one converted label, and each wedge represents one original label from source data.

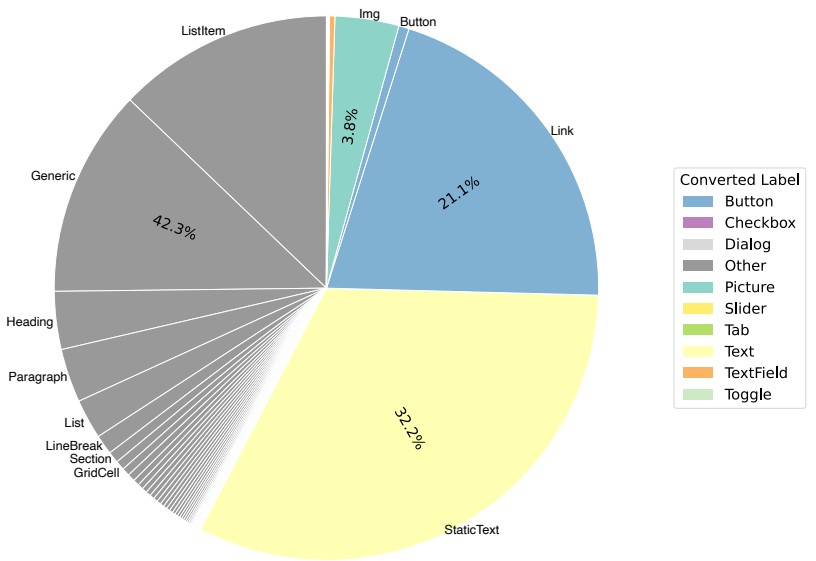

Figure 10: The statistics of original labels (**wedge**) and converted labels (**legend**) on webpage platform. Each color represents one converted label, and each wedge represents one original label from source data.

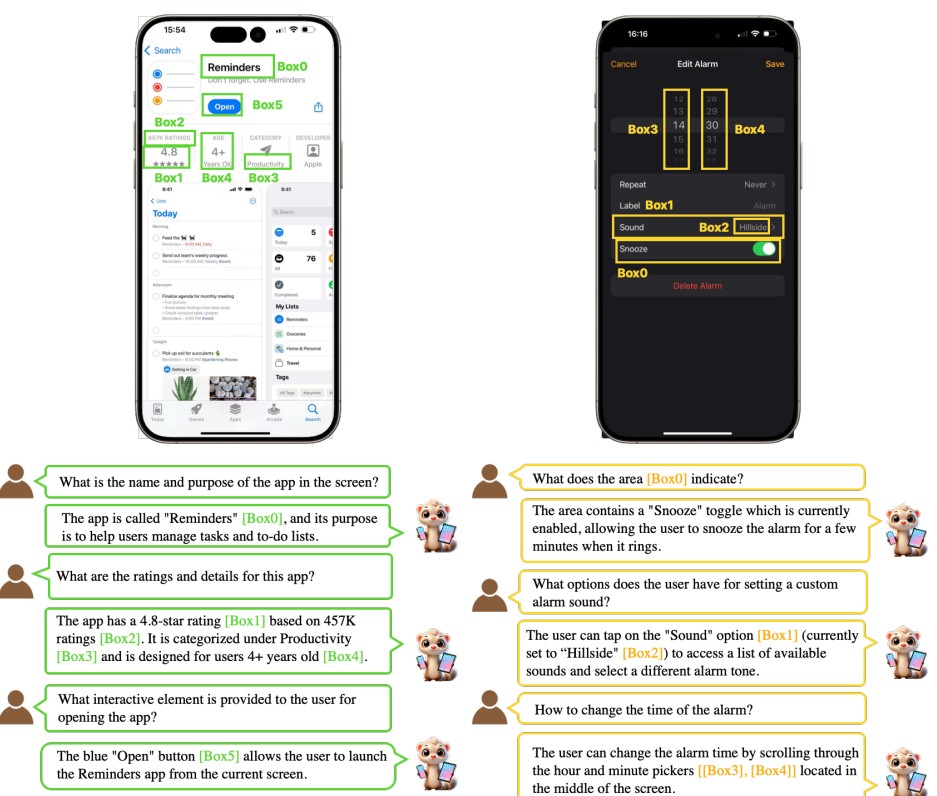

Figure 11: Real examples of the Ferret-UI 2 model interacting with iPhone.

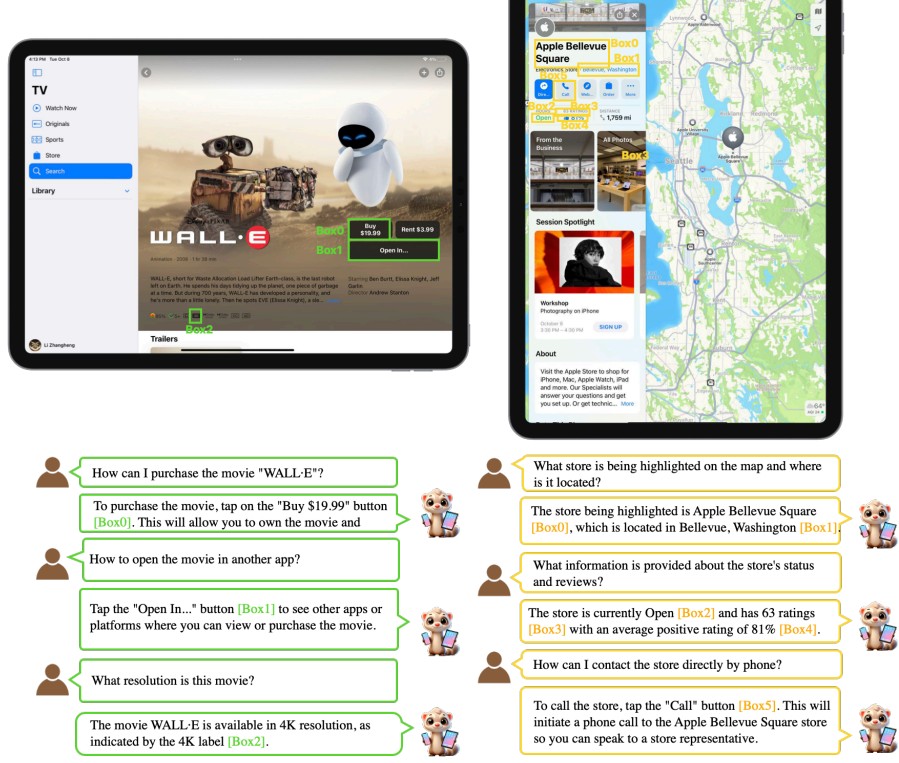

Figure 12: Real examples of the Ferret-UI 2 model interacting with iPad.

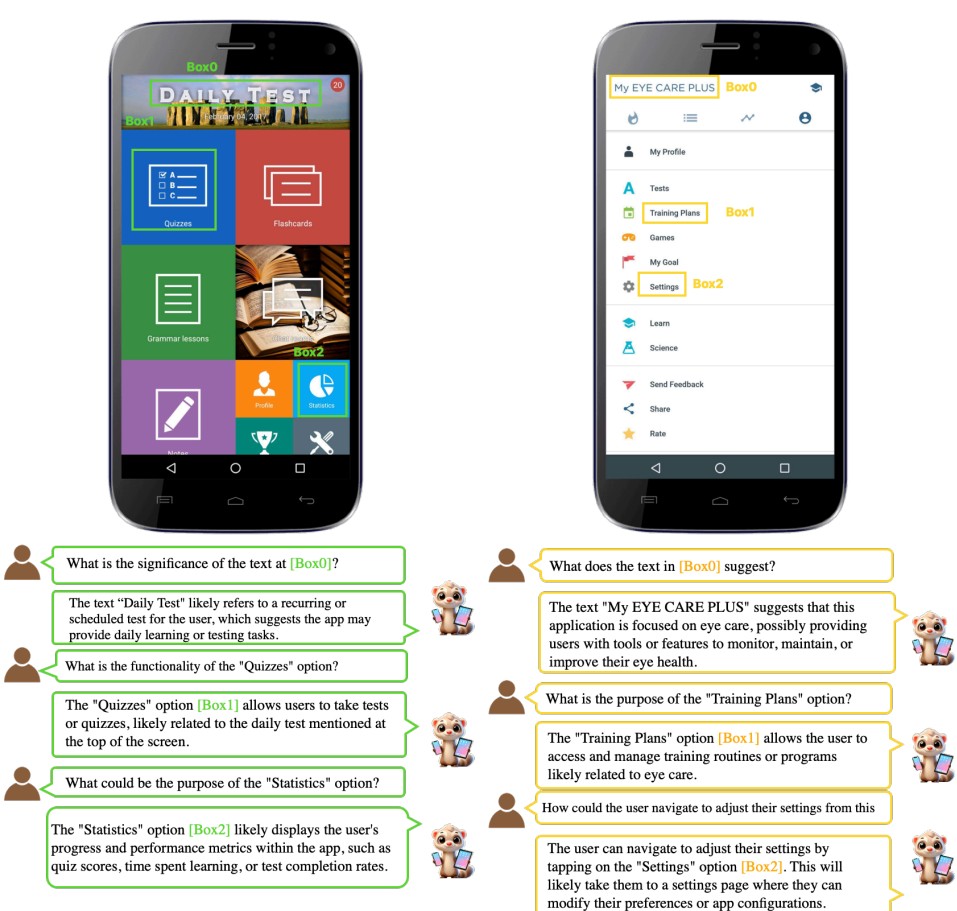

Figure 13: Real examples of the Ferret-UI 2 model interacting with Android.

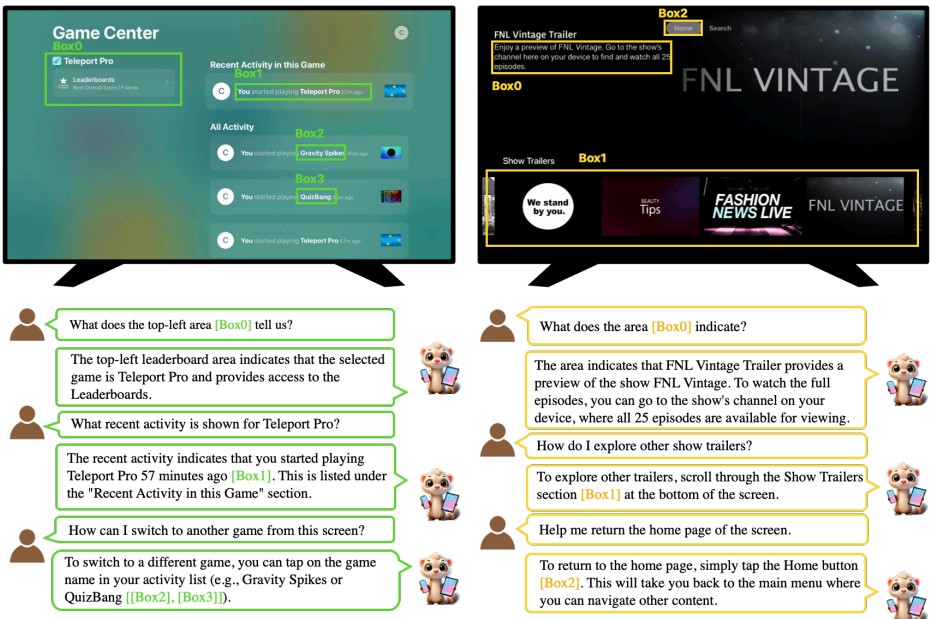

Figure 14: Real examples of the Ferret-UI 2 model interacting with AppleTV.

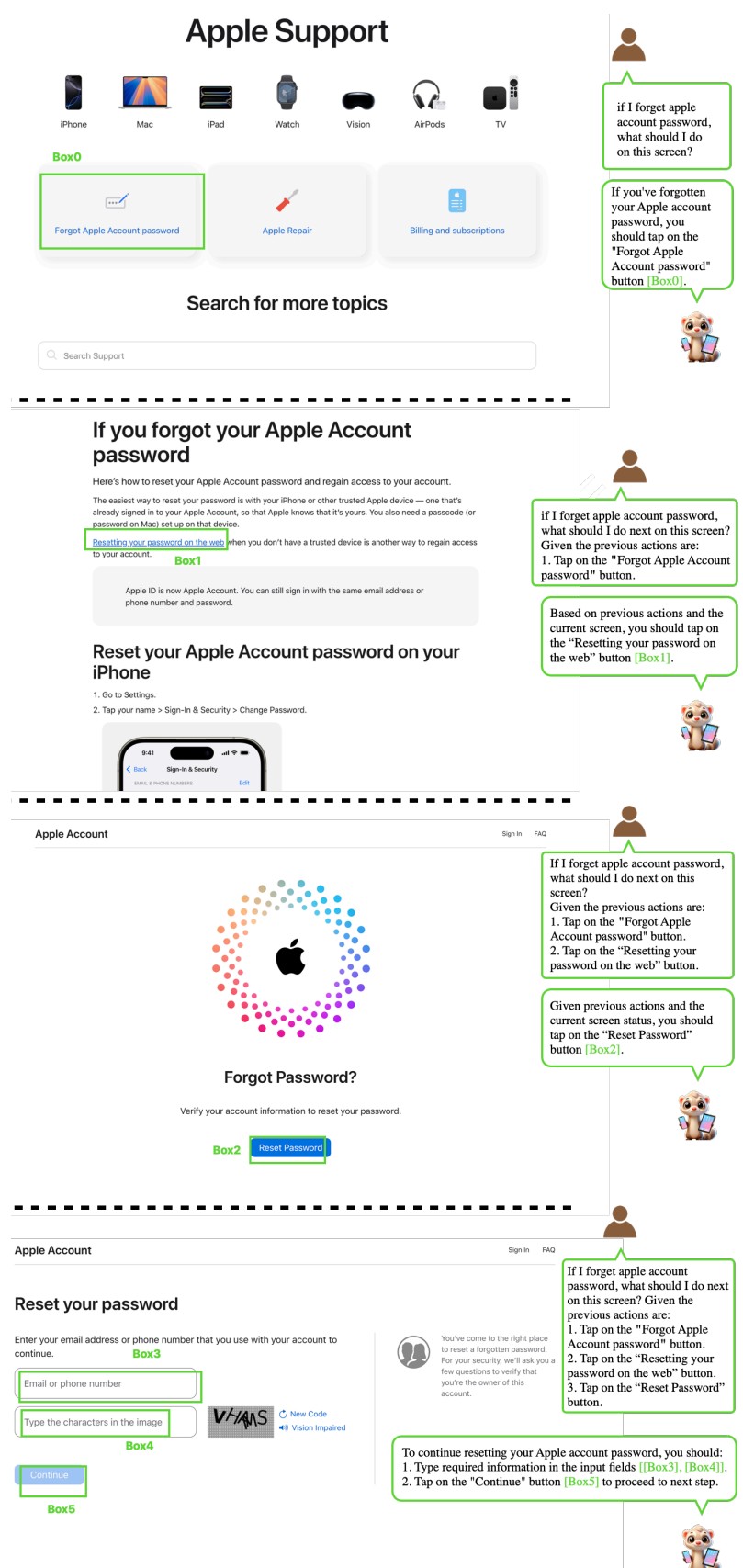

Figure 15: An example of the Ferret-UI 2 model performing multi-step interactions on real-time webpages following GUIDE-style QAs.

