# OpenReview forum: "Ferret-UI 2: Mastering Universal User Interface Understanding Across Platforms"
_ICLR.cc/2025/Conference — ICLR 2025 Poster_

### Official Review · Reviewer_u2Px · 2024-10-19

**Soundness:** 2
**Presentation:** 3
**Contribution:** 3
**Rating:** 6
**Confidence:** 5

**Summary:**

The paper introduces an MLLM-based GUI Agent aimed at enhancing user interface (UI) understanding across a wide range of platforms. These platforms include mobile devices (iPhone, Android), tablets (iPad), web pages, and smart TVs (AppleTV). The paper builds on previous work, Ferret-UI, proving its efficacy on referring, grounding, and advanced user-centric tasks, showing significant improvements over the previous Ferret-UI model and other benchmarks like GUIDE and GUI-World.

**Strengths:**

- **Multi-platform Support**: One of the paper's key strengths is the successful integration of multiple platforms (iOS, Android, web, smart TVs), making the model more versatile in various user environments. The authors also discuss the transfer learning effects of each platform.
- **Advanced Task Training with GPT-4o**: The use of GPT-4o with visual prompting for training data generation significantly enhances the model’s performance, particularly on complex tasks requiring spatial understanding of UI components.
- **Strong Empirical Results**: The results presented in the paper indicate that Ferret-UI One outperforms its predecessor, Ferret-UI, across several metrics, including grounding, referring, and advanced task handling, particularly on the GUIDE and GUI-World benchmarks.
- **Zero-shot Transfer**: The model demonstrates robust zero-shot performance, indicating that it generalizes well across different platforms without overfitting to training data, a key advantage for building scalable UI systems.

**Weaknesses:**

- **Extra Probing for Models Potential**: This paper mainly follows the pipeline of GUI-world, proposing a multi-platform GUI agent by collecting the dataset first and annotating it with SOTA MLLM GPT-4o. Although its performance is SOTA in GUI-World and GUIDE benchmark, the whole paper is not that exciting to me.  I think your model's experiment results are expected to be superior to Ferret-UI because Ferret-UI One is trained on more high-quality instruction tuning and task-specific data. The authors are suggested to probe more potential capability of their models such as the grounding plugin for general MLLM-based GUI Agents in benchmarks such as OS-world or VisualAgentBench.
- **Evaluation and Analysis of Synthetic Dataset**: The authors are suggested to evaluate whether GPT-4o can perform the annotation task well by providing some quantitative results. Moreover, a comprehensive comparison to previous GUI datasets and analysis of your dataset is needed.
- **Case Study**: Show more case studies in the appendix and provide failure cases of Ferret-UI One in GUIDE and GUI-World in grounding and navigation tasks for a more comprehensive review of your proposed model, as well as an analysis of why your model fails. A more detailed discussion of future direction is also suggested on topics of building more general agents across various platforms or how to improve the grounding or navigating capabilities of current models or agents.
- **Additional Experiments**:  I think authors can include more comparative baselines to show the strength of your synthetic data generation pipeline. The authors can consider SeeClick and SeeClick-V on your proposed benchmark, two strong baseline models in REC, grounding, and navigation tasks because these models are all recently released and show high potential in GUI grounding tasks. Another option is to evaluate your model in popular GUI grounding benchmark ScreenSpot proposed in SeeClick.

**Questions:**

See Weakness Above.

**Details Of Ethics Concerns:**

The dataset is collected in various GUI platform, which may contain user's privacy or copyright problem for content in images. Therefore, I suggest the authors to add a discussion section for potential copyright problem and select the most suitable license for your proposed dataset.

---

> ### Author Response · Authors · 2024-11-24
> **Author response**
>
> We thank the reviewer for the detailed feedback and constructive suggestion. Below, we address key concerns and provide clarifications with additional experiments and analyses.
>
> >**Q1:  Extra Probing for Model's Potential:** The experiment results are expected to be superior to Ferret-UI because Ferret-UI One is trained on more high-quality instruction tuning and task-specific data.
>
> **A1:** Our contributions go beyond merely engineering improvements. While we acknowledge that Ferret-UI One builds upon its previous model, we introduce new scientific methodologies to overcome specific challenges in cross-platform UI understanding. Key advancements include:
>
> -   Adaptive N-gridding for enhanced high-resolution perception, which optimally balances local and global visual encoding across devices. This method allows Ferret-UI One to accurately perceive widgets on screens of various resolutions while minimizing distortion. Table 5 demonstrates the benefits of this technique.
>
> - Closing the domain gap among different platforms: Another major challenge is the multi-platform integration of GUI understanding. We believe one of our contributions is finding that through curated data processing and uniformation, we can mitigate the domain gaps of GUI understanding across different platforms and successfully train a single model that well understands UI knowledge from different platforms.
>
> >**Q2:** Provide grounding test results on OS-world and VisualAgentBench.
>
> **A2**: We want to clarify that our data pipeline is built based on Ferret-UI. We appreciate the suggestion to explore additional benchmarks such as OS-world and VisualAgentBench. However, our primary goal in this work was to demonstrate Ferret-UI One's effectiveness and generalizability across a diverse set of platforms and benchmarks, such as GUI-World, which is a comprehensive and challenging benchmark for multi-platform UI understanding tasks. The GUIDE is also a challenging benchmark for probing multi-step navigation potential under single-step settings by providing action history.\
> While OS-world and VisualAgentBench are valuable benchmarks, they focus on broader or slightly different aspects of UI interaction and may not align with the specific goals of our study. Specifically, VisualAgentBench emphasizes on real-time and multi-step navigation in simulation environments, while OS-world focuses on desktop which requires much broader knowledge of operating systems like Windows, Ubuntu and MacOS and is beyond our scope. We believe our chosen benchmarks adequately capture the core capabilities of Ferret-UI One, such as fine-grained grounding, referring, and robust cross-platform transfer.\
> We agree that the superior performance of Ferret-UI One compared to its predecessor, Ferret-UI, is partially due to high-quality instruction tuning and task-specific data. However, it is important to emphasize that Ferret-UI One also introduces significant innovations, such as adaptive high-resolution gridding and enhanced multi-platform training data generation, which contribute to its strong results. These advancements were specifically designed to address challenges inherent in multi-platform GUI understanding and generalization.\
> To further explore the model's potential, we have conducted additional analyses within the scope of our existing benchmarks. For instance, we include detailed case studies of grounding and navigation tasks, analyze failure cases, and evaluate cross-platform transferability. These analyses provide deeper insights into the model's performance and potential limitations, which we believe are more relevant and impactful for the current scope of the paper.
>
> >**Q3:  Evaluation and Analysis of the Synthetic Dataset**: Provide quantitative analysis of GPT-4o's annotation quality and compare it with previous datasets.
>
> **A3:** We conducted a quantitative evaluation of GPT-4o annotations by manually checking a randomly sampled subset consisting of 100 advanced task examples. The result shows that GPT-4o achieved an accuracy of 86% in terms of choosing correct bounding boxes from visual prompting, and achieved a 4.6/5 avg score on the quality of the textual answer.

---

> ### Author Response · Authors · 2024-11-24
> **Author response**
>
> >**Q4:** Provide a comprehensive comparison to previous GUI datasets and analysis of your dataset is needed.
>
> **A4:** We provide a comprehensive comparison to previous multi-platform datasets.
>
>
> | Dataset         | # Sample | Platforms                               | Tasks                                                                                   |
> |-----------------|----------|-----------------------------------------|-----------------------------------------------------------------------------------------|
> | OmniAct         | 9,802    | Desktop, Web                           | Code Generation                                                                         |
> | OS-World        | 369      | Desktop, Web                           | General Control                                                                         |
> | AITW            | 715,142  | Android(Apps+Web)                      | Navigation                                                                              |
> | GUI-world       | 12,379   | iPhone, Android, Web, XR, Desktop       | GUI understanding, instruction following                                               |
> | Ferret-UI       | 123,702  | iPhone, Android                        | **Elementary**: Referring, Grounding.  **Advanced**: Function/detailed description, perception and interaction QA |
> | Ferret-UI One   | 529,411  | iPhone, Android, iPad, AppleTV, Web     | **Elementary**: Referring, Grounding.  **Advanced**: Comprehensive description, perception and interaction QA       |
>
> In addition, we discuss in detail and provide flowgram comparison on how the data pipeline of Ferret-UI is different from ours in Appendix C. Importantly, we highlight following key differences between Ferret-UI One and Ferret-UI:
>
> - Multi-platform support: The training data of Ferret-UI One consists of 5 platforms compared to 2 platforms of Ferret-UI.
>
> - Raw annotation qualities: The majority part of bounding boxes, labels and on-screen text annotations of Ferret-UI One is either extracted from source data or annotated by human, while these annotations of Ferret-UI are all generated from model detection.
>
> - Bounding box prompting: When generating advanced tasks, Ferret-UI One uses GPT-4o+SoM visual prompting for bounding boxes, while Ferret-UI uses purely textual prompting containing coordinates, bbox labels and text annotations.
>
> - Advanced task quality: Due to the constraint of textual prompting, Ferret-UI pipeline cannot perceive the visual content of UI elements, thus limiting the quality and content diversity of generated advanced tasks.
>
>
>
> >**Q5: Case Studies and Failure Analysis:** Include failure cases.
>
> **A5:** We have added a section analyzing typical failure cases of Ferret-UI One in referring, grounding and navigation tasks in Appendix I. Typical referring and grounding failures include recognizing rendered text in images, wrongly recognizing widget-like objects contained in images, grounding widgets with blurry edges, requiring to performing too many navigation steps in one screenshot, and unseen navigation patterns in training set.
>
> >**Q6:** Discuss limitations
>
> **A6:** (1) One major limitation of our current dataset is the simplified action space. Currently we mainly support some simple actions: tapping/clicking, typing and scrolling etc. More advanced actions, such as multi-finger swiping and zoom-in/out gestures are currently not supported. (2) Multi-lingual support: our model currently supports English-only GUI understanding. (3) Noise exists in advanced tasks generated by GPT-4o+SoM visual prompting when the bounding boxes are dense in the given screenshot, causing 14% average error rate when GPT-4o tries to refer to the correct bounding boxes. Finding efficient ways to eliminate noises in such large scale synthetic data is urgently needed.
>
>
> >**Q7:** A more detailed discussion of future direction is also suggested on topics of building more general agents across various platforms or how to improve the grounding or navigating capabilities of current models or agents.
>
> **A7:** We discuss this in rebuttal to **Q3** for reviewer #**zXTq**. To summarize, in order to make state-of-the-art UI agents useful in practice, extensive future work is necessary in (1). Improving the agents’ general completion rate. (2). Improving the agents' self-reflection and replanning capabilities. (3). A careful study of safety concerns.

---

> > ### Author Response · Authors · 2024-11-24
> > **Author reponse**
> >
> > >**Q8: Additional Experiments**: The authors can consider SeeClick and SeeClick-V on your proposed benchmark ... Another option is to evaluate your model in popular GUI grounding benchmark ScreenSpot proposed in SeeClick.
> >
> > **A8:** We evaluate the SeeClick model pre-trained on their original data on our proposed benchmarks and GUIDE benchmark and additionally evaluate the pre-trained Ferret-UI One model on Screenspot benchmark. The results on the GUI-World benchmark demonstrate that Ferret-UI One outperforms SeeClick in zero-shot testing.
> >
> > |      Model      |   Backbone    | Refer  | Ground  | Adv-score | Adv-IoU | GUIDE-score | GUIDE-IoU | GUI-World |
> > |:---------------:|:-------------:|:------:|:-------:|:---------:|:-------:|:-----------:|:---------:|:---------:|
> > |   Ferret-UI     | Vicuna-13B    |  64.15 |  57.22  |   45.81   |  18.75  |    41.15    |   26.91   |     -     |
> > | Ferret-UI One   | LLAMA3-8B     | **81.34** | **81.31** | **86.25** | **41.71** | **88.81**   | **54.71** |   2.948   |
> > |     GPT-4o      |      -        |  56.47 |  12.14  |   77.73   |   7.06  |    75.31    |    9.64   | **3.619** |
> > |    SeeClick     | Qwen-VL-9.6B  |  51.58 |  62.82  |   67.49   |  21.56  |    54.70    |   39.51   |   2.704   |
> >
> >
> > For SeeClick-V model, we were unable to find an available resource for testing.
> >
> > On Screenspot benchmark, the Ferret-UI One achieves 54.0% average accuracy, outperforming CogAgent and SeeClick, and in particular, achieving good performance on our supported Mobile and Web platforms while achieving fair performance on the unseen desktop platform.
> >
> > |     Model      | Model Size | Mobile Text | Mobile Icon/Widget | Desktop Text | Desktop Icon/Widget | Web Text | Web Icon/Widget | Avg   |
> > |:--------------:|:----------:|:-----------:|:------------------:|:------------:|:--------------------:|:--------:|:---------------:|:-----:|
> > |    CogAgent    |     8B     |    67.0%    |       24.0%        |    **74.2%**     |        20.0%         |   70.4%  |      28.6%      | 47.4% |
> > |   SeeClick     |   9.6B     |    78.0%    |       52.0%        |    72.2%     |        **30.0%**         |   55.7%  |      32.5%      | 53.4% |
> > | Ferret-UI One  |     8B     |   **80.3%** |      **55.4%**     |   52.1%  |       21.7%      | **81.2%**|     **33.5%**   | **54.0%** |
> >
> >
> > >**Q9: Ethical Concerns:** I suggest the authors to add a discussion section for potential copyright problem and select the most suitable license for your proposed dataset.
> >
> > **A9:** Thanks for your kind consideration. Our proposed dataset is collected internally and underwent comprehensive legal reviews before usage. We will add a proper license if the data is to be released in the future.

---

> > > ### Comment · Reviewer_u2Px · 2024-11-24
> > > **Thanks for your rebuttal**
> > >
> > > Thanks for the response and additional experiments. I do acknowledge this work is a good follow-up of Ferret-UI and GUI-World that incorporates various platforms. I think now the paper is above my acceptance threshold and have raised my rating

---

### Official Review · Reviewer_EWX6 · 2024-10-23

**Soundness:** 3
**Presentation:** 4
**Contribution:** 3
**Rating:** 8
**Confidence:** 4

**Summary:**

Edit: Updated Review based on Feedback from Associate Program Chairs

This paper introduces Ferret-UI One, a multimodal large language model designed for universal UI understanding across multiple platforms including iPhone, Android, iPad, Web, and AppleTV. The key innovations are threefold: multi-platform support through unified data collection and processing, high-resolution perception through an adaptive N-gridding mechanism, and advanced task training data generation using GPT-4o with set-of-mark visual prompting. The authors demonstrate the model's effectiveness through comprehensive evaluations on elementary tasks, advanced tasks, and public benchmarks like GUIDE and GUI-World.

**Strengths:**

The paper's technical contributions are well-motivated. The adaptive N-gridding mechanism provides an elegant solution to handling varying resolutions while maintaining bounded computational costs, with clear mathematical formulation in Algorithm 1. The set-of-mark visual prompting approach for GPT-4o data generation represents an acceptable way to improve spatial understanding in training data generation, addressing a key limitation of previous text-only prompting methods.
The authors conduct extensive experiments across multiple platforms and task types, providing strong baseline comparisons including GPT-4o and previous state-of-the-art models. The ablation studies examining cross-platform transfer capabilities and architectural components offer valuable insights into the model's behavior. Particularly noteworthy is the model's strong performance even with smaller variants like Gemma-2B, suggesting practical applicability.
The paper addresses a genuine need in the field for universal UI understanding across diverse platforms. The demonstrated zero-shot generalization capabilities, especially in the GUI-World benchmark results, indicate robust real-world applicability.

**Weaknesses:**

A significant concern is the severe data imbalance across platforms, as evident in Table 1. The web platform dominates with 321k images, followed by iPhone with 112k images, while iPad and AppleTV have only 19k and 16k images respectively. Though the authors acknowledge this limitation and mention mitigation strategies like loss weighting and targeted task generation, the effectiveness of these approaches isn't thoroughly analyzed or quantified. To mitigate this, the answer is in balancing the data used across devices.
The methodological analysis has several gaps. The paper lacks a detailed examination of failure cases or systematic error analysis. There's limited discussion of computational requirements and inference speed comparisons across different platforms and resolutions. (for model it'll be specific, but in UI design, speed is a major element and latency numbers there would be helpful). Inference speed and overall latency in milliseconds is be a good metric. Additionally, potential biases in GPT-4o generated training data aren't thoroughly explored (although I consider it to be out of scope of this paper), which could impact the model's generalization capabilities.
The experimental design could be more rigorous. The cross-platform transfer results presented in Table 4 would benefit from error bars / error spread measures. Wherever appropriate, provide confidence intervals / standard deviations and running the experiment multiple times to ensure reliability.

**Questions:**

2. Can you provide detailed examples of failure cases, especially for low-resource platforms like AppleTV?
3. How does the quality of GPT-4o generated training data vary across platforms, and are there systematic differences in generation quality?
4. What are the inference time comparisons between platforms for Ferret-UI 1 ?

**Details Of Ethics Concerns:**

No ethical issues.

---

> ### Author Response · Authors · 2024-11-24
> **Author response**
>
> We thank the reviewer for the thoughtful feedback and for acknowledging the strengths of our work. Below, we can address your concerns as follows:
>
> >**Q1:** A significant concern is the severe data imbalance across platforms, as evident in Table 1.
>
> **A1:** We acknowledge the data imbalance, particularly between high-resource platforms (e.g., web, iPhone) and low-resource platforms (e.g., iPad, AppleTV). To mitigate this, we applied targeted approaches:
>
> -   **Loss Weighting:** During training, we assigned higher loss weights to lower-resource platforms like iPad and AppleTV based on their scale, encouraging the model to focus on these underrepresented data. We tried using balanced data to train the model, by cropping the data amount for platforms with excessive data, but found it not as effective as simply using the loss weighting to balance the training. Specifically, when balancing the example numbers from each platform to 66k, we find that the average performance is 1.5% lower than training with full scale data and loss weighting where the weight is inversely proportional to the task data scale.
>
> -   **Advanced Task Generation:** We supplemented low-resource platforms with additional advanced tasks generated from existing examples, using diverse prompts and varying visual contexts. In particular, for AppleTV and iPad data, for each screenshot, we generate data for all the 3 advanced tasks which equals to 3 training examples, while for other platforms we randomly generate data for only one advanced task for each screenshot equaling to 1 training example.
>
>
> >**Q2:**  Can you provide detailed examples of failure cases, especially for low-resource platforms like AppleTV?
>
> **A2:** We have added a section analyzing typical failure cases of Ferret-UI One in referring, grounding, and navigation tasks in Appendix I. Typical referring and grounding failures include recognizing rendered text in images, wrongly recognizing widget-like objects contained in images, grounding widgets with blurry edges, requiring performing too many navigation steps in one screenshot, and unseen navigation patterns in the training set.
>
> Besides failure case study, we have also added new visual examples of our model across diverse platforms in the Appendix for visually understanding the capabilities of our model.
>
>
> >**Q3:** How does the quality of GPT-4o generated training data vary across platforms, and are there systematic differences in generation quality?
>
> **A3:** We conducted a quantitative evaluation of GPT-4o annotations by manually checking a randomly sampled subset consisting of 100 examples, where each sample could be either advanced task we proposed. The result shows that GPT-4o achieved an accuracy of 85% in terms of choosing correct bounding boxes from visual prompting and achieved a 4.6 out of 5 human evaluation score on the quality of the textual answer. Since our cross-platform raw data is highly uniformed before advanced task generation with GPT-4o, we do not notice significant systematic differences in generation quality.
>
> >**Q4:** What are the inference time comparisons between platforms for Ferret-UI 1 ?
>
> **A4:** We evaluate the inference speed of our model with different sizes in single-step inference setting(i.e. One screenshot and one QA each time), tested on one A100GPU with 500 samples. Result shows that our model with Gemma2B backbone uses 0.89s/frame, while the model with Llama8b backbone uses 1.29s/frame, and model with Vicuna-13B consumes 1.45s/frame. Since Ferret-UI One shares the same codebase with Ferret-UI, which only uses Vicuna-13B as backbone, we obtain a similar inference speed of 1.45s/frame for Ferret-UI.
>
> >**Q5:** The experimental design could be more rigorous. The cross-platform transfer results presented in Table 4 would benefit from error bars / error spread measures.
>
> **A5:** We argue that it is not feasible to include error bars in Table 4 since it is a zero-shot test setting. In zero-shot testing, the model's performance is assessed without fine-tuning or training on the target task. As a result, there is no inherent variance from retraining or reinitializing the model, which typically contributes to error bars.

---

> ### Comment · Reviewer_EWX6 · 2024-11-24
> **Response to Authors:**
>
> I appreciate the authors' detailed response addressing my concerns. The quantitative analysis showing the 1.5% performance trade-off with loss weighting versus balanced data sampling provides valuable insight into their design choices. The detailed inference timing benchmarks across different model sizes (0.89s/frame for Gemma2B to 1.45s/frame for Vicuna-13B) and the thorough failure case analysis in Appendix I add significant value to the study. I am raising my rating to 8 based on these clarifications.

---

### Official Review · Reviewer_zXTq · 2024-11-03

**Soundness:** 3
**Presentation:** 3
**Contribution:** 2
**Rating:** 5
**Confidence:** 3

**Summary:**

The authors present a model to recognize widgets on device screens, an evolution of a previously-published (in arxiv) model by Apple. The model performs better than the previous model, and slightly better than GPT-4o.

**Strengths:**

The paper is well written and fairly easy to understand. It provides a clear comparison with the previous version, what simplifies the reader's task of understanding what is novel.

The proposed model seems to perform significantly better than the previous one, in spite of greater generality. Its performance is comparable, if not slightly better, than GPT-4o.

**Weaknesses:**

Since the authors fail to disclose whether the model is going to be available, and how, it is hard to evaluate the impact and relevance for the community of the work described here.

Most importantly, since most of the improvements are engineering-related, the main take-away of the paper is that Ferret-UI, a paper which was never peer-reviewed, can be improved. In other words, there is not much novelty here, beyond knowing it can be done, since the methods used are quite straight-forward.

Notice that the performance of the model is probably still too low to make it useful. It fails elementary and ground tasks at the rate of 1 and 5, that is, in most screens it will be incorrect in relation to at least one widget. This is probably too much for any serious attempt to connect it to an agent, at least in a safe way. In fact, a more interesting and useful benchmark is how successful such models are in actually interacting with a screen, but this is beyond this paper.

**Questions:**

1. What do you think are the scientific contributions of this paper? How much of it is just good engineering of tested ideas?
2. How is this model going to be made available to researchers and developers?
3. What would you consider a "usable" performance of a model like Ferret-UI?

**Details Of Ethics Concerns:**

I think it is easy to expect that the authors of this paper are mostly from Apple, given that this is an evolution of a published model.

They could have done a much better of anonymization (for instance, calling it with a different name), and I wonder whether this is an issue here.

For full disclosure, I do not think I have a significant conflict of interests with Apple in this space.

---

> ### Author Response · Authors · 2024-11-24
> **Author response**
>
> We thank the reviewer for the thoughtful feedback and for acknowledging the strengths of our work, particularly the model's clarity, writing quality, and performance over prior work. Below, we respond to specific points raised in the review.
>
> >**Q1:** What do you think are the scientific contributions of this paper? How much of it is just good engineering of tested ideas?
>
> **A1:** Our contributions go beyond merely engineering improvements. While we acknowledge that Ferret-UI One builds upon its previous model, we introduce new scientific methodologies to overcome specific challenges in cross-platform UI understanding. Key advancements include:
>
> - **Adaptive N-gridding:** Enhancing high-resolution perception, which optimally balances local and global visual encoding across devices. This method allows Ferret-UI One to accurately perceive widgets on screens of various resolutions while minimizing distortion.
>
> - **Closing the domain gap among different platforms:** Another major challenge is the multi-platform integration of GUI understanding. We believe one of our contributions is finding that through curated data processing and uniformation, we can mitigate the domain gaps of GUI understanding across different platforms and successfully train a single model that well understands UI knowledge from different platforms. Specifically,
> 	- Table 2 shows significant improvements over previous methods (89.73 GPT-4o score vs 45.81 for Ferret-UI)
> 	- Table 3 demonstrates strong zero-shot performance on GUI-World benchmark
> 	- Table 4 shows systematic analysis of zero-shot transfer capabilities between platforms. The results reveal important patterns about UI knowledge transfer:
> Strong transfer between iPhone/iPad/Android (>65% performance);
> Limited transfer from AppleTV to mobile platforms;
> Platform similarity in resolution and aspect ratio impacts transfer success.
>
> While the paper does include significant engineering work, the core scientific contributions around adaptive N-gridding and cross-platform learning advance our theoretical understanding of UI comprehension, which hits a balance between novel insights and engineering implementation.
> We also argue that its predecessor model, namely the Ferret-UI, has been peer-reviewed and published in ECCV 2024 [1].
>
> >**Q2:** Since the authors fail to disclose whether the model is going to be available, and how, it is hard to evaluate the impact and relevance for the community of the work described here.
>
> **A2:**
> We are actively exploring avenues for sharing Ferret-UI One with the research community while ensuring compliance with legal and organizational constraints. Particularly, its predecessor models, Ferret-UI, and relevant code have already been made available on Github, and we also plan to release our model.

---

> ### Author Response · Authors · 2024-11-24
> **Author response**
>
> >**Q3:** What would you consider a "usable" performance of a model like Ferret-UI?
>
> **A3:** We argue that all state-of-the-art UI agents show a low completion rate in UI navigation, which requires significant research and improvements to make them useful in practice.
>
> To address your concern, we’ve built a UI agent on top of Ferret UI models for multi-step navigation, similar to CogAgent [2]. The agent can interact with UI screens to perform actions including tapping, swiping, and typing. In our preliminary study on multi-step navigation, we design a set of 100 online test episodes from across 20 first party apps and third party apps, all were evaluated on real iOS devices. Episodes covered a spectrum of use cases (e.g., managing notes in the Notes app, modifying a contact in the Contacts app, navigating in Maps, creating a calendar invite). To measure an agent's navigation performance accurately and repeatably, we build an online evaluation framework with careful staging and evaluation.
>
> We evaluate our UI agent against the GPT4o agent with set of mark prompting [3] by task completion rate. A navigation result is rated by three independent annotators, and the averaged results are shown in the following table. Our UI agent achieves a 53.2% completion rate and outperform GPT4 agent by 10%. This shows the great potential of building UI agents on top of strong UI understanding models. On the other hand, we also acknowledge that the achieved 53.2% completion rate is not satisfactory enough. Similar low completion rate has also been found in WebArena (WebRL[4]), with only 40% completion rate.
>
> | Agent model        | Task completion rate |
> |:------------------:|:--------------------:|
> | GPT4o-SoM Agent    | 43.7%               |
> | Ferre-UI One Agent | 53.2%               |
>
> Though out-of-the-scope of the Ferret UI One paper, we conducted an error analysis on several agent models on UI navigation tasks, including state-of-the-art CogAgent [2], GPT4-SoM agent [3], and the agent model built on top of Ferret UI. We identified that mistakes in planning, rather than action grounding, account for the majority of navigation failures in all models, a similar conclusion can be found in UIGround [5] as well. Specifically, all agents lack replanning and self-reflection capabilities in challenging scenarios, which require in-depth app exploration. Hallucinations may occur when finding hidden UI elements.
>
> Besides, the agents may make dangerous mistakes, such as deleting contacts, or leaving unexpected comments. Developing a holistic guardrail to prevent agents from risky actions is an important future step for safe UI navigation, which has been only recently explored in a preliminary study [6].
>
> To summarize, in order to make state-of-the-art UI agents useful in practice, extensive future work is necessary in (1) improving the agents’ general completion rate; (2) improving the agents' self-reflection and replanning capabilities; (3) a careful study of safety concerns.
> What’s more, Ferret-UI One is not a model specifically made for navigation but focuses on understanding and single-step interaction. Besides navigation, the model can also be used in other applications such as accessibility, automated testing and so on.
>
> >**Q4:** Ethics Concerns
>
> **A4:** We would like to argue that our name choice is driven by technical claity rather than the affiliation. Furthermore, our paper stands out disregarding its original as discussed in the scientific contribution point. There are many papers built on previous work with the same conventions, such as LLaVA-1.5[7] and SAM 2[8].
>
> **References:**
>
> [1] You, Keen, et al. "Ferret-ui: Grounded mobile ui understanding with multimodal llms." European Conference on Computer Vision. Springer, Cham, 2025.
>
> [2] Hong, Wenyi, Weihan Wang, Qingsong Lv, Jiazheng Xu, Wenmeng Yu, Junhui Ji, Yan Wang et al. "Cogagent: A visual language model for gui agents." In Proceedings of the IEEE/CVF Conference on Computer Vision and Pattern Recognition, pp. 14281-14290. 2024.
>
> [3] Yan, An, et al. "Gpt-4v in wonderland: Large multimodal models for zero-shot smartphone gui navigation." arXiv preprint arXiv:2311.07562 (2023).
>
> [4] Qi, Zehan, et al. "WebRL: Training LLM Web Agents via Self-Evolving Online Curriculum Reinforcement Learning." arXiv preprint arXiv:2411.02337 (2024).
>
> [5] Gou, Boyu, et al. "Navigating the digital world as humans do: Universal visual grounding for gui agents." arXiv preprint arXiv:2410.05243 (2024).
>
> [6] Zhang, Zhuohao Jerry, et al. "From interaction to impact: Towards safer ai agents through understanding and evaluating ui operation impacts." arXiv preprint arXiv:2410.09006 (2024).
>
> [7] Liu, Haotian, et al. "Improved baselines with visual instruction tuning." _Proceedings of the IEEE/CVF Conference on Computer Vision and Pattern Recognition_. 2024.
>
> [8] Ravi, Nikhila, et al. "Sam 2: Segment anything in images and videos." _arXiv preprint arXiv:2408.00714_ (2024).

---

> ### Comment · Reviewer_zXTq · 2024-11-25
>
> Based on the answers from the authors, I still think that this work is mostly engineering improvement of the previous, non-reviewed work.
>
> It seems that the authors agree with my assessment that the current accuracy is not enough to make it useful, what is not said in the paper. I am not sure whether the authors are planning to do so and, by doing it, lowering their claims.
>
> It is good to know that they are planning to make it available.
>
> Considering that it is mostly an engineering paper which provides incremental gains but not useful results, I believe my rating is appropriate.

---

> > ### Author Response · Authors · 2024-11-26
> > **Further reponse**
> >
> > We thank the reviewer for their continued feedback and for acknowledging our plans to make the model available to the community. We address the remaining concerns in detail below.
> >
> > ----------
> >
> > **1. Ferret-UI Was Peer-Reviewed**
> >
> > We would like to clarify again that _Ferret-UI_, the predecessor of _Ferret-UI One_, was peer-reviewed and published at ECCV 2024. Please refer to this link:
> > https://eccv.ecva.net/virtual/2024/poster/749
> >
> > ----------
> >
> > **2. On Usability and Incremental Progress**
> >
> > We respectfully disagree with the characterization that the current navigation accuracy renders our work "not useful". First, our work focuses on UI understanding, and Ferret-UI One is the SOTA UI understanding model as evidenced by the empirical results presented. Second, it is essential to understand that advancing GUI agents is a multi-step process, and achieving our ultimate goal—a highly reliable and universally applicable UI agent—requires tackling numerous intermediate challenges, and each milestone contributes meaningfully toward long-term progress.
> >
> > Our advancements address key bottlenecks in the GUI agent field——multi-platform UI understanding—— as also identified in [9]. Specifically, UI grounding and understanding remain the primary bottleneck for building reliable UI agents capable of complex navigation and interactions. Without robust UI comprehension, higher-level functionalities like planning and safe execution cannot succeed.
> >
> > We also want to emphasize again that a strong UI understanding model is not only useful for GUI agent navigation, but also in other UI applications such as accessibility and automated testing.
> >
> > ----------
> >
> > **3. Addressing Current and Future Impact**
> >
> > The reviewer notes: "It seems that the authors agree with my assessment that the current accuracy is not enough to make it useful, what is not said in the paper."
> >
> > To clarify, we do not view the current state of _Ferret-UI One_ as insufficient but rather as part of an iterative process to address long-standing challenges in UI understanding. The pursuit of general-purpose UI agents is not a single-step endeavor. As evidenced by related works like UGround [5], WebRL [4], CogAgent[2], and many others, all SOTA agents face limitations in accuracy and reliability. These limitations do not negate their contributions but instead highlight areas where further research is needed.
> >
> > Moreover, the paper explicitly discusses failure cases and opportunities for improvement, emphasizing transparency about the current state of the model. However, this transparency does not diminish the paper's claims but reinforces its role in the ongoing progress toward solving these challenges.
> >
> > ----------
> >
> > **4. Contributions Beyond Engineering**
> >
> > While this work incorporates engineering improvements, it also advances the scientific understanding of multi-platform UI comprehension. Contributions such as adaptive N-gridding and systematic analyses of cross-platform transferability provide valuable insights for the research community. These methods are not mere extensions of existing ideas but represent new approaches to addressing specific challenges in UI understanding, as demonstrated by significant improvements in performance metrics like GUIDE IoU and GUI-World zero-shot benchmarks.
> >
> > Furthermore, we also want to emphasize that novelty does not necessarily mean creating something entirely new. Enhancing existing methods to advance the current state of knowledge, conducting careful engineering and comprehensive empirical study to draw new insights should also be qualified as contributions and is beneficial to the community.
> >
> > We hope this response clarifies our position and reaffirms the value of _Ferret-UI One_.
> >
> > **Reference**
> >
> > [9] Gou B, Wang R, Zheng B, et al. Navigating the digital world as humans do: Universal visual grounding for gui agents[J]. arXiv preprint arXiv:2410.05243, 2024.

---

> > > ### Author Response · Authors · 2024-12-02
> > > **Follow-up**
> > >
> > > Dear reviewer,  we want to follow up regarding our response to your review. If there are any additional concerns or further clarifications needed, we’d be more than happy to provide additional information. We look forward to hearing from you.

---

### Author Response · Authors · 2024-11-24
**Paper revision summary**

Dear AC and reviewers:

We thank all the reviewers for their detailed and valuable feedbacks and suggestions. Our revised paper has been uploaded. To summarize, we have following major updates in the revised version:

Addressing Reviewers' Concerns:
 1. Include failure cases and analyses in Appendix I.
 2. A comprehensive dataset comparison with prior work in Appendix C.
 3. Add SeeClick test results on our benchmarks in Table 2 and 3.
 4. Add Screenspot benchmark results in Appendix E.

What's new:
 1. We provide more inference examples in Appendix F, in particular a successful multi-step navigation example.
 2. Resolution statistics of our dataset in Appendix G.
 3. Label statistics and uniform mapping results of raw data from different platforms in Appendix H.

We hope our response can utmostly address your concerns!

---

### Meta-Review · Area_Chair_CCrn · 2024-12-21

**Metareview:**

This paper introduces an MLLM designed for universal user interface understanding. It is verified on multiple platforms and achieved better performance than previous methods and the GPT-4o. The major concerns are about public availability, novelty, practical usability, and more experiments. After rebuttal, the authors addressed most of the concerns, and two reviewers improved their scores. Reviewer zXTq remains a negative score with a relatively low confidence score due to the engineering improvement over the previous unreviewed paper and practical usability issues. The authors clarified their novelty, the publication information of the previous paper, and usability issues. Though the reviewer did not provide further comments, from the AC's view, the concerns have mostly been addressed. Considering the overall rating and the additional information in the discussion, the final recommendation is accept.

**Additional Comments On Reviewer Discussion:**

This paper received three reviews.
Reviewer zXTq's concerns were about the availability, novelty, and practical usage issues. The authors provided detailed responses, and the reviewer insists on the initial rating of 5.
Reviewer EWX6's concerns were the data imbalance, failure case analysis, inference speed comparison, and GPT-4o bias issues. The author's responses addressed these concerns, and the review improved the score from 6 to 8.
Reviewer u2Px's concerns were the novelty, analysis with previous datasets and baselines, and more case studies. The author provided new experimental results to help address these concerns, and the reviewer improved the score from 5 to 6.
Finally, Reviewer zXTq remains the concern and provides a negative rating with low confidence. Considering all the reviews and discussions, the AC finds these concerns could be well addressed. Therefore, the final recommendation is an acceptance.

---

### Decision · Program_Chairs · 2025-01-22

Accept (Poster)